# Resource selection of a nomadic ungulate in a dynamic landscape

**Theresa S. M. Stratmann**[1,2]*, **Nandintsetseg Dejid**[2], **Justin M. Calabrese**[3], **William F. Fagan**[4], **Christen H. Fleming**[4,5], **Kirk A. Olson**[6], **Thomas Mueller**[1,2]

**1** Department of Biological Sciences, Goethe University, Frankfurt am Main, Germany, **2** Senckenberg Biodiversity and Climate Research Centre, Senckenberg Gesellschaft für Naturforschung, Frankfurt am Main, Germany, **3** Center for Advanced Systems Understanding, Görlitz, Germany, **4** Department of Biology, University of Maryland, College Park, Maryland, United States of America, **5** Smithsonian Conservation Biology Institute, National Zoological Park, Front Royal, Virginia, United States of America, **6** Mongolia Program, Wildlife Conservation Society, Ulaanbaatar, Mongolia

* theresastrat@gmail.com

**Data Availability Statement:** For conservation concerns it is not standard to publish the raw GPS data on animal locations, but we provide a data frame with re-centered GPS points, all absence points, and all associated environmental data. This

## Abstract

Nomadic movements are often a consequence of unpredictable resource dynamics. However, how nomadic ungulates select dynamic resources is still understudied. Here we examined resource selection of nomadic Mongolian gazelles (*Procapra gutturosa*) in the Eastern Steppe of Mongolia. We used daily GPS locations of 33 gazelles tracked up to 3.5 years. We examined selection for forage during the growing season using the Normalized Difference Vegetation Index (NDVI). In winter we examined selection for snow cover which mediates access to forage and drinking water. We studied selection at the population level using resource selection functions (RSFs) as well as on the individual level using step-selection functions (SSFs) at varying spatio-temporal scales from 1 to 10 days. Results from the population and the individual level analyses differed. At the population level we found selection for higher than average NDVI during the growing season. This may indicate selection for areas with more forage cover within the arid steppe landscape. In winter, gazelles selected for intermediate snow cover, which may indicate preference for areas which offer some snow for hydration but not so much as to hinder movement. At the individual level, in both seasons and across scales, we were not able to detect selection in the majority of individuals, but selection was similar to that seen in the RSFs for those individuals showing selection. Difficulty in finding selection with SSFs may indicate that Mongolian gazelles are using a random search strategy to find forage in a landscape with large, homogeneous areas of vegetation. The combination of random searches and landscape characteristics could therefore obscure results at the fine scale of SSFs. The significant results on the broader scale used for the population level RSF highlight that, although individuals show uncoordinated movement trajectories, they ultimately select for similar vegetation and snow cover.

data set is sufficient to run the statistical analyses and reproduce the figures and is located in Open Science Framework (https://osf.io/w6syz/) at: DOI 10.17605/OSF.IO/W6SYZ.

**Funding:** TSMS was funded by the MainCampus Stipendium der Stiftung Polytechnischen Gesellschaft Frankfurt am Main. TSMS, TM, and ND were funded by the Robert Bosch Foundation. TSMS, TM, and ND were funded by the German Federal Ministry of Education and Research (BMBF, 01LC1710A and 01LC1820A). JMC was partially funded by the Center of Advanced Systems Understanding (CASUS) which is financed by Germany's Federal Ministry of Education and Research (BMBF) and by the Saxon Ministry for Science, Culture and Tourism (SMWK) with tax funds on the basis of the budget approved by the Saxon State Parliament. JMC, CHF, and WFF were supported by NSF Grant IIBR 1915347. The funders had no role in study design, data collection and analysis, decision to publish, or preparation of the manuscript.

**Competing interests:** The authors have declared that no competing interests exist.

# Introduction

In landscapes where resource availability is dynamic, animals often migrate [1]. When resources are both dynamic and unpredictable, start and end points for a migration no longer exist, and animals must respond more flexibly, often through nomadic movements [2–4]. While much is known about resource selection of migratory animals—those animals which move between fixed locations at predictable times—much less is known about resource selection of nomadic animals. The studies that do exist are often confined to one or two seasons within a year. Yet drivers of movement may differ between seasons, especially in temperate zones when snow can alter resource selection [5–7]. To complicate matters, nomadic animals show great individual variability in movement trajectories within the same landscape, a pattern that contrasts with the behavior of many migratory animals that show movement patterns which are more consistent across individuals [4]. Whether this individual variation is a result of among individual differences in resource selection, or simply due to random searches for food, remains unknown. To examine drivers of resource selection across the year within nomadic animals we used Mongolian gazelles (*Procapra gutturosa*) for this case study.

## Resource selection during the vegetation growing season

For ungulates the forage maturation hypothesis (FMH) describes how plant characteristics and herbivore foraging constraints might interact to influence herbivores' selection for vegetation. The FMH states that herbivores should select for vegetation of intermediate biomass [8–13]. New vegetation is highly nutritious but low in quantity and thus time consuming to find and eat. Older vegetation is high in quantity, requiring little time to find and eat, but low in nutritional value. This is due to plants building up their cell walls with lignin and ramping up chemical defenses as they grow, making them more difficult for herbivores to digest. Selecting for intermediate biomass therefore optimizes energy intake. Hindgut fermenters can better meet their energy requirements on lower quality vegetation than ruminants. Ruminants require time for their efficient digestion, whereas hindguts can compensate for low quality vegetation by eating large quantities that are quickly digested [14]. We would therefore expect the FMH to be especially relevant for a ruminant like a Mongolian gazelle [14]. The FMH was further developed into the Green Wave Hypothesis (GWH) [15–19] which adds a spatial and temporal component to the FHM. It hypothesizes that animals migrate over elevational or latitudinal gradients to take advantage of multiple peaks in forage quality. However, in landscapes such as range- and drylands, in which nomadic ungulates search for unpredictable forage, the GWH may not apply, because green wave tracking works best when the gradient of vegetation quality lasts over a long time and moves in a predictable order rapidly across the landscape [20, 21]. This is why in this manuscript we have focused on the FMH. Elk (*Cervus elaphus*) [13, 22], wildebeest (*Connochaetes taurinus*) [23], and Thomson's gazelles (*Gazella thomsoni thomsoni*) [24] have all been found to select intermediate vegetation biomass according to the FMH. Two studies have even found support for nomadic ungulates following the FMH, i.e., nomadic animals seem to be able to find locations with high quality forage while searching an unpredictably changing landscape [24, 25]. For example, [24] found that nomadic Thompson's gazelles positioned themselves in areas of above-average energy gain during the wet season. In addition, a previous study on Mongolian gazelles [25] found evidence for the FMH based on presence-absence data from transect surveys.

## Resource selection in winter

While the FMH and GWH describe how ungulates are expected to select for forage during the growing season, selection strategies must shift during winter. For ungulates in temperate

climates cold temperatures and snow become key factors shaping resource selection. Snow can increase the cost of movement, restrict access to forage, or even prevent movement altogether [6, 26]. On the other hand, snow can serve as a means to hydrate [27, 28]. Cold temperatures and icy winds can also introduce trade-offs between selecting for habitat types that offer shelter from thermoregulatory stress and deep snow (e.g., dense forest) and selecting for those that provide good forage (e.g., open meadows) [7, 29–31]. While selection for habitat types differs by species, most studies on winter habitat selection of ungulates find avoidance of deep snow, especially in small ungulates [5, 7, 26]. For ungulates living along elevation or latitudinal gradients this means moving to the valleys or lower latitudes [5, 32]. For ungulates in open steppes snow fall is less predictable, so the ability to move to escape snow storms is likely important, especially since there are no forests to seek cover in [33]. Because forage quality is generally low in winter, resource selection should be shaped by the need to maintain access to forage and the need to keep energetic costs low. While there are several examples of studies of resource selection of ungulates in winter, few of these focus on nomadic ungulates. Two previous studies have examined population level resource selection of Mongolian gazelles in winter and both found selection for high forage biomass and areas with a short snow cover duration or little snow cover [28, 34].

## Importance of examining individual selection

Past studies of resource selection have often focused on selection at the population level, that is, on the average individual [35–37]. These studies pooled data or results across individuals to arrive at an overall conclusion about what resources a population selects for. Yet recently studies have found individual differences in habitat selection which is forcing researchers to consider the implications of these differences on management practices [35, 36]. Considering individual selection and examining differences among individuals is especially interesting when studying nomadic ungulates because individuals often show uncoordinated movements with greatly varying trajectories [4, 38]. The cause of these uncoordinated movements is poorly understood and difficult to study. Potential explanations are that individuals use a random search strategy to cope with unpredictable resource dynamics or that individuals show varying selection strategies due to different physiological states. Yet to-date no one has looked at individual variation in resource selection of Mongolian gazelles.

## Life history of Mongolian gazelles

Mongolian gazelles are small, approximately 30kg ungulates native to the steppes of Russia, Mongolia, and China. Most Mongolian gazelles occur in the Eastern Steppe of Mongolia, the area east of the Ulaanbaatar—Beijing railroad, north of the Gobi Desert, and south of the Siberian forests. It is estimated that there are around 800,000 gazelles [39] making them a species of least concern [40]. Mongolian gazelles are intermediate feeders, exploiting high quality grasses and forbs during the growing season, and surviving off of dead vegetation in the winter by increasing the surface area of the rumen [41, 42]. This change in physiology is key to surviving in this highly seasonal environment where green vegetation is mainly present from June to September and the winter is long and harsh with temperatures dropping below -40˚C. Unpredictability in vegetation dynamics and snowfall are thought to be key drivers of this species' nomadic movements.

Mongolian gazelles are considered nomadic based on previous quantitative analyses of GPS location data [3, 38, 43–45]. Fleming et al. [44] showed that it takes a gazelle several months to cross the area it uses during its lifetime, which can be up to three-times as large as the Serengeti-Mara ecosystem. It has also been shown that individual Mongolian gazelles do not return

to specific locations, not even during calving time or winter [38, 43, 45]. The large area used by gazelles, the long time it takes to cross that area, and the lack of repeatability in their movement are the key characteristics of their nomadic movement. Group dynamics are another important part of the movement behavior of gazelles. Individuals will move with conspecifics in herds, but they will leave one herd and join another in a fission-fusion manner [*sensu* 46, also see video in supporting materials for 38]. This means that individuals which are found in the same herd on one date can have completely different movement paths in the following days to years [3, 38, 43]. Average annual range size for Mongolian gazelles is ~ 19,346 km$^2$ but individual annual range sizes differ by an order of magnitude (range 6,431 km$^2$–53,422 km$^2$) [38]. This individuality in movement motivated us to examine not just population, but also individual responses to vegetation and snow cover.

### Research approach and hypotheses

To study resource selection of Mongolian gazelles we used an extensive data set of daily GPS locations of 33 individuals. During the growing season we associated this data with the Normalized Difference Vegetation Index (NDVI), a satellite based measure of vegetation greenness that has been linked to vegetation quality and has been used to quantify resource availability [47–50]. Usually studies use field based measures (e.g., biomass, nitrogen content) to test the FMH. However, given the size of the Eastern Steppe and the temporal duration of our study, such field measurements would not have been feasible. Few other studies have used NDVI to test the FMH [25, 51]. However, since high NDVI values correspond to higher vegetation biomass [52–55] we believe it is justified to test the FMH using NDVI. According to the FMH herbivores should avoid high biomass vegetation, which corresponds to high NDVI values. They should also avoid low biomass vegetation which corresponds to low NDVI values. Therefore, the FMH would predict selection for intermediate NDVI [25, 56]. Since Mongolian gazelles are ruminants, we hypothesized that they should select for intermediate NDVI during the growing season. In winter, foraging is mediated by snow depth, which can prevent movement and access to forage for small ungulates. On the other hand, snow is required as a source of drinking water. Since data on snow depth is not available for Mongolia, we used MODIS snow cover to examine resource selection in winter. We therefore hypothesized that gazelles should select for areas with intermediate snow cover during the winter, providing access to both water and forage. Finally, we were interested in comparing population and individual level selection and hypothesized that selection would be similar at the population and at the individual-level. Despite the unique paths individual Mongolian gazelles take, we expected that these different paths would nonetheless lead Mongolian gazelles to areas with similar characteristics because there is a unifying physiological need for high quality vegetation and areas with low snow cover. To test these hypotheses, we (i) used resource selection functions (RSFs) [57] to examine population-level selection within the steppe for NDVI during the growing season and snow cover in winter and (ii) step-selection functions (SSFs) [58] to examine individual selection for NDVI and snow cover along the movement path, at 1, 5, and 10 day step-lengths.

## Methods

### Study area

The Eastern Steppe of Mongolia is one of the largest intact temperate grasslands left in the world [42]. It is an area with open plains, rolling hills, few standing or flowing sources of water, and no trees. There are only a few human settlements and few paved roads in this area, although oil and mineral extraction is becoming more common. Fences are rare in this landscape, except 1) at the border to Russia and China and 2) on both sides of the railroad leading

from Ulaanbaatar to Beijing, both of which are highly effective at stopping gazelle movement [38, 59].

The location of good pastures for wild and domestic herbivores is unpredictable, changing based on rain, snow, and fire events [60–62], eliciting nomadic movements in both humans and Mongolian gazelles. Mean annual rainfall is around 200 mm [62] and temperatures range from over 30˚C in the summer to -40˚C in the winter [27]. It is thought that this harsh climate controls both wild and domestic herbivore populations more than forage limitation does [63]. Mongolian gazelles must therefore be able to tolerate a wide range of conditions across the year.

## Mongolian gazelle tracking data

Mongolian gazelles were captured in the Eastern Steppe of Mongolia using drive nets [64]. This method of capture follows a standard protocol that was approved by the Ministry of Environment and Green Development in Mongolia (licenses: 6/5621, 5/4275). In October of 2014 16 Mongolian gazelles were captured. Fifteen of these Mongolian gazelles were outfitted with Lotek GPS collars programmed to collect location data every 23 hours, and one Mongolian gazelle was outfitted with a solar-powered Sirtrack GPS collar programmed to collect location data every hour. Two Mongolian gazelles died within a month of capture and so their data were not used. In September of 2015, 19 Mongolian gazelles were captured. Fourteen Mongolian gazelles received the Lotek GPS collars collecting daily locations and 5 received the solar-powered Sirtrack GPS collars collecting hourly data. Seven of the 33 Mongolian gazelles were male and all individuals were adults. We analyzed data up to May 8, 2018, giving us over 3.5 years of data for some individuals (range: 22 to 1286 days, $\mu \pm sd$ = 411.5±315.1 days, S1 Table). Data from hourly Mongolian gazelles were summarized as a daily average location, so all data analyses were based on daily location data. The GPS tracks of the gazelles within the Eastern Steppe can be seen in Fig 1.

## Remote sensing data

We used NDVI to quantify resource availability during the vegetation growing season and snow cover to quantify resource availability during winter. For the growing season, we used the 250 meter, 16 day NDVI data (MOD13Q1) from Terra MODIS (version 6) and downloaded it from the LP DAAC (https://lpdaac.usgs.gov/) and LAADS (https://ladsweb.modaps.eosdis.nasa.gov/) databases using the MODIS package in R [65]. To quantify snow cover we used the 500 meter resolution data set on maximum snow cover extent during an 8 day period (MOD10A2) from Terra MODIS (version 6). We used the MODIS package to download the data from the National Snow and Ice Data Center (https://nsidc.org/data/). In this categorical snow data set, pixels are classified as: snow (if snow was detected at least once over the 8 days), no snow, missing data, no decision, night, lake, ocean, cloud, lake ice, detector saturated, or fill. Using ESRI ArcGIS 10.6 we reclassified the data set into three categories: snow, no snow, and put the other categories into a no data category, as they provide no information on snow cover. We then used the aggregate tool to create a data set of mean snow cover extent in a 2 x 2km area. We chose to create this coarser resolution data set because snow can affect mobility and we were interested in not just the snow cover at the recorded GPS point, but what the surrounding area was like. While snow depth would be the most informative measure to use, such a data set does not exist for the Eastern Steppe.

## Defining seasons

We split our analyses into two seasons, the growing season and winter. Seasons were defined based on the presence of snow cover because it alters the interpretation of NDVI. To define

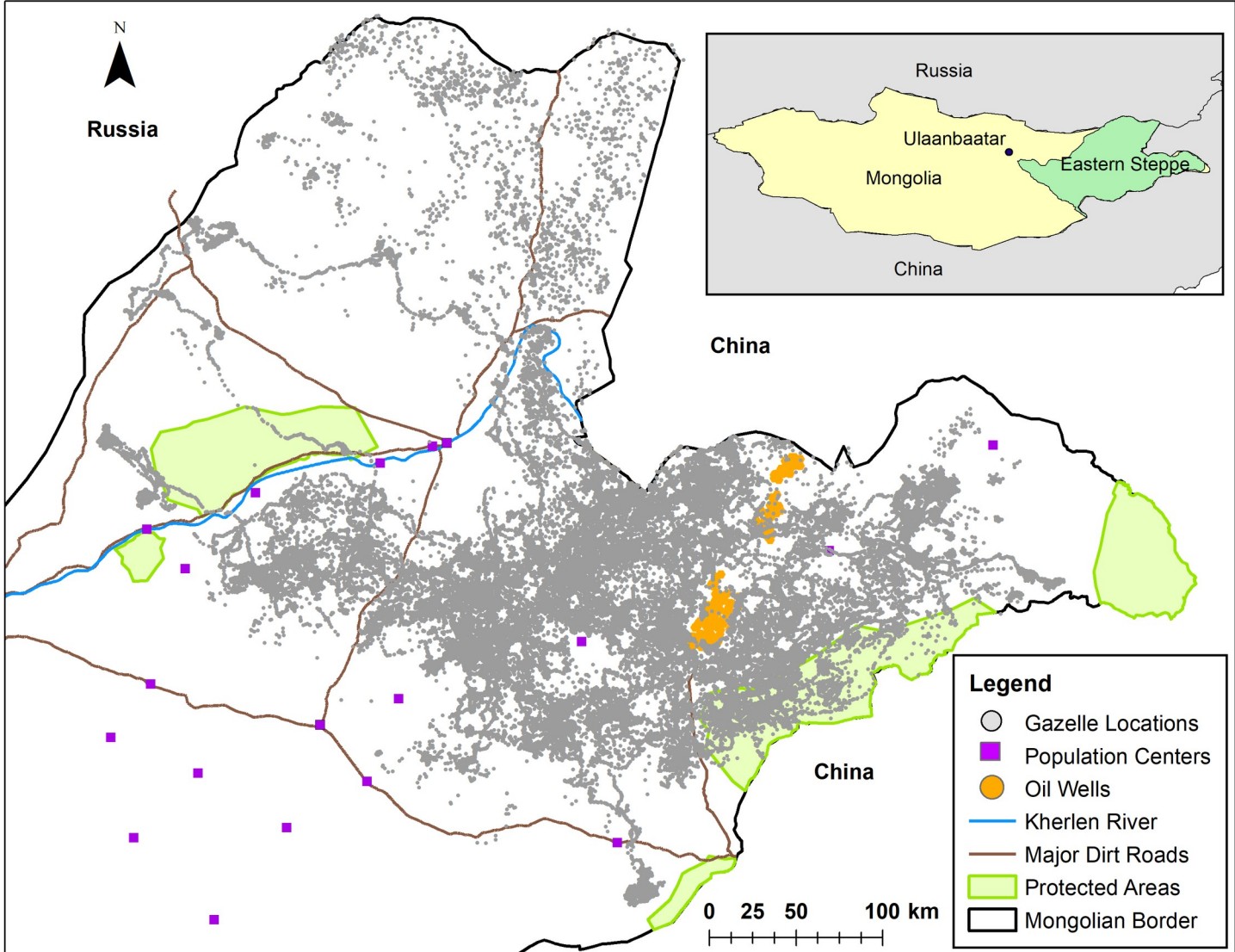

**Fig 1. Map of the study area and GPS locations of 33 Mongolian gazelles within the Eastern Steppe of Mongolia.** The inset map shows the portion of the Eastern Steppe that is east of the fenced railroad that goes from Ulaanbaatar to Beijing. Note how the fences along the border prevent Mongolian gazelles from entering into Russia and China.

the growing season, we chose only dates for which no snow cover was recorded during the years 2014 to 2018, based on the snow cover data set. This timeframe was from May 9 to August 29. The remainder of the year, from August 30 to May 8, we call winter since it is the period when snow can fall. While we could have examined more than two periods by taking the gazelle's life history into account (e.g., calving season during the growing season and rutting season during winter) these periods would have been too short to have sufficient data.

## Resource selection functions

All data preparation and analyses for the RSFs and SSFs were conducted in R (R version 3.5.1 and 4.0.1, accessed 10 July 2018 and 22 June 2020) [66].

To examine resource selection at the population-level we ran generalized linear mixed-effects models (GLMMs) for each season, pooling the daily GPS points from all Mongolian

gazelles. This analysis therefore reflects what the average Mongolian gazelle selects for within the Eastern Steppe. To conduct this analysis we generated a set of 'pseudo-absence' points to use as a comparison for the visited points. To confine our pseudo-absence points spatially we created a 100% minimum convex polygon (MCP) around all daily points for each individual-season-year combination using the package adehabitatHR [67]. We then cropped the MCPs to be within Mongolia since Mongolia is fenced and gazelles cannot cross into Russia or China (Fig 1). While the 100% MCPs include some areas the Mongolian gazelles did not visit, their high mobility would have nonetheless made these areas available to them. The 100% MCPs also allowed us to examine how Mongolian gazelles placed themselves within the larger landscape as a contrast to the SSFs which looked at selection along the movement path. Our models were run on a seasonal basis and each season included data from several MODIS NDVI scenes (n = 7 per year) or MODIS snow cover scenes (n = 32 per year), which summarize NDVI over a 16 day period and snow cover over an 8 day period. The coarser temporal resolution compensates for days on which data quality is poor due to cloud cover. RSFs require a large number of random points [68] so we created data sets with 1, 2, 5, 10, and 20 times more pseudo-absence points than presence points. This allowed us to determine when model beta coefficients became stable. We split the environmental data for our pseudo-absence points evenly among the MODIS scenes. To do this, for each MODIS scene/individual gazelle combination we: 1) selected the MCP for that individual-season-year, 2) used the spsample function from the R package sp [69, 70] to generate a stratified random sample of pseudo-absence points within the MCP equal to (or 2, 5, 10, or 20 times more than) the number of presence points falling within that MODIS scene, 3) extracted the NDVI or snow cover values from that scene to the random points, and 4) repeated the procedure for each scene-individual combination. We then used the glmer function in the lme4 package [71] to run the GLMMs. To avoid biased estimates of the beta coefficients we gave presence points a weight of 1 and pseudo-absence points a weight of 1000 [72]. For each season, we ran two models. The first model had a quadratic term for the fixed effect of either NDVI (e.g., $NDVI + NDVI^2$) or snow cover with random slopes and intercepts for each individual Mongolian gazelle and random intercepts for year and MODIS scene. A quadratic model can test whether individuals are selecting for intermediate resource values (downward parabola) or both the minimum and maximum available resource values (upward parabola). The second model had a linear term for the fixed effect, NDVI or snow cover, with random slopes and intercepts for each individual Mongolian gazelle and random intercepts for year and MODIS scene. This linear model tests whether Mongolian gazelles selected for the maximum resource value (positive slope) or the minimum resource value (negative slope). For each season we then used AIC to compare support for these hypotheses regarding the shape of selection (i.e., quadratic vs. linear). In addition to testing two models, we ran all models with an increasing number of pseudo-absence points. While we tried to account for non-independence in our data (e.g. data coming from the same individual, year, or MODIS scene) with random effects, these models often resulted in singular fits, indicating that our model was too complex for the data [71]. We therefore had to reduce model complexity in several cases. In the growing season, the linear and quadratic models were singular because year did not explain any variance in the data. We therefore re-ran the models without year as a random intercept. Models were still singular, because individual also explained little variance, and so we ran the models with only a random intercept for MODIS scene. In winter, the random intercept for MODIS scene resulted in singular fits in both the linear and quadratic models, so we re-ran the models without a random intercept for MODIS scene. In the linear, but not quadratic, models, the random intercept for year caused a singular fit and so was removed.

## Step selection functions

To examine selection of individual Mongolian gazelles for NDVI and snow cover along the movement path we used step-selection functions (SSFs). SSFs use conditional logistic regression (clogit) to compare areas where Mongolian gazelles were found to areas where the animals could have gone, based on their step-lengths and relative turn angles [58]. SSFs group comparisons of a presence point and its corresponding pseudo-absence points into one step. The pseudo-absence points represent locations that the Mongolian gazelle could have also reached in a given time step but did not go to. For each step the environment at the presence location is compared with the environment at the pseudo-absence locations. Doing this for many steps allowed us to determine if there was a significant difference in the environment between where a Mongolian gazelle was found and where it could have been. SSFs allow us to examine selection along the movement path.

Since resource selection is scale dependent [73] the biological interpretation of SSFs depends on the interval between locations (the step-length) [74]. It is unknown at what scale Mongolian gazelles select for vegetation and snow cover. We therefore tested multiple scales. Usually, the vegetation (or snow cover) a day's walk away (~ 5 km displacement on average; S1 Fig) is very similar. We therefore also tested step lengths of 5 days (~17 km displacement on average; S1 Fig) and 10 days (~ 27 km displacement; S1 Fig). This allowed us to sample a variety of spatial scales, but because sample size diminishes with each increase in step-length we were not able to test larger steps. To generate the data sets for the different scales we went through all the daily locations identifying sets of consecutive GPS locations with a minimum of three points per set. Three points is the minimum required to calculate a relative turn angle. We did the same for points 5 and 10 days apart. From these data we could then calculate 1, 5, and 10 day displacements (e.g., step-lengths) and relative turn angles. Data from all individuals were pooled to lessen the influence of unusual displacement lengths or low sample sizes for individuals with less data. Using these pooled data of step-lengths and turn angles, we created 25, 50, 100, 200, and 400 pseudo-absence points for each point visited by the animal. NDVI or snow cover values were then extracted to the presence and pseudo-absence points. The appropriate MODIS scene to use was chosen by matching the date of the presence point with the MODIS scene that included that date.

SSFs were run on an individual-season-year basis (e.g., Mongolian gazelle 1 in the winter of 2015) for all three spatial scales (1, 5, and 10 day step-lengths) for the winter models because mixed-effects models such as those proposed by [37] usually did not converge for our data set. Yet for the growing season models the 5 and 10 day step scales had too little data to split it by year, we therefore ran these models by individual and ran the 1 day models by both individual-year and individual as comparison. We used the 'clogit' function in the survival package [75, 76] to run our models. We only kept individual-season-year or individual-season combinations that had at least 20 steps. For each individual-season-year data set we fit a linear model (just NDVI or just snow cover) and a quadratic model (NDVI + NDVI$^2$ or snow cover + snow cover$^2$) to test the same hypotheses we tested in the RSFs. Within each season-year, we determined how many individuals showed significant selection ($\alpha \leq 0.05$) in either the linear or quadratic models. We used AIC to determine which hypothesis had more support, but we do not report on top models that were not significant as our hypotheses likely did not capture the underlying process well in these cases. Based on the top models we categorized selection into selection for the maximum (linear model with a positive slope), selection for the minimum (linear model with a negative slope), selection for intermediate values (quadratic model, downward parabola), or selection for extreme values (quadratic model, upward parabola). Since there is no good rule of thumb for the number of required pseudo-absences, models were run

with 25, 50, 100, 200, and 400 pseudo-absence points to examine when beta coefficients became stable.

To visualize our data we plotted relative selection strength against available NDVI or snow cover [77]. Relative selection strength acknowledges that selection is always dependent on a choice between two or more options. Because we were interested in the FMH we chose mean NDVI for a given year as our comparison point for the growing season models by individual-year. For the models by individual, we chose mean NDVI during the 2015–2017 growing seasons. For the winter models we used mean snow cover for a given year because we also hypothesized that gazelles would select for intermediate snow cover. The mean value was calculated by pooling both the presence and pseudo-absence data within a season from all individuals, but dividing it by year for the yearly means. Because these values show some extremes due to a few points falling in very rare habitats, we restricted the data by removing the upper and lower 5% of NDVI (snow cover) values. The relative selection strength curves show selection strength for NDVI (snow cover) values available to an individual in a given season-year if that individual must choose between those values or the mean NDVI (snow cover) value. Available NDVI (snow cover) is the range of NDVI (snow cover) covered by the points the individual Mongolian gazelle visited and by the associated pseudo-absence points. Again, we restricted the available values by removing the upper and lower 5% of NDVI (snow cover) values. A characteristic of the plotting method we used is that different reference points result in different selection strengths, but the NDVI (snow cover) value at which selection peaks stays the same. For comparisons between individuals, across seasons and years we therefore focused on the NDVI (or snow cover) value at peak selection and not selection strength.

## Results

For the RSF models we ran all models with an increasing number of pseudo-absence points. Coefficients did not change much after using 20 times the number of pseudo-absence points as presence points so we present results from these models (S2 Fig). At the population level, in both seasons, quadratic models offered a better fit than linear models (Table 1). During the growing season gazelles selected for intermediate NDVI (Fig 2; $\beta_{NDVI}$ = 13.34, $p < 0.001$; $\beta_{NDVI}^2$ = -14.55, $p < 0.001$; S2 Table). The selected NDVI value (0.46, the NDVI value at which log(RSS) peaks) was above the average NDVI value within the population range over the three growing seasons (0.30) (Fig 2). In fact, the selected value was at the upper end of the NDVI available to gazelles (0.15–0.55). We defined available NDVI to exclude NDVI values in the upper and lower 5% of the data. Compared to the maximum and minimum NDVI

**Table 1. AIC values for the different growing season (May 9 –Aug. 29) and winter (Aug. 30 –May 8) RSF models examining how Mongolian gazelles select for NDVI and snow cover.**

| Season | Hypothesis | df | AIC | Weight | ΔAIC |
|---|---|---|---|---|---|
| Growing Season | Quadratic | 4 | 73515 | 1.00 | 0.0 |
| | Linear | 3 | 73592 | <0.001 | 77.1 |
| Winter | Quadratic | 7 | 209339 | 1.00 | 0.0 |
| | Linear | 4 | 209389 | <0.001 | 50.7 |

Models represent different hypotheses on the shape of selection (linear or quadratic) for NDVI during the growing season and for snow cover during winter. During the growing season, the models have a random intercept for NDVI scene, but no random slope for individual or random intercepts for individual and year because even simplified versions of these random effects resulted in singular fits. For winter, the quadratic model has a random slope for individual and random intercepts for individual and year. Adding an intercept for MODIS scene resulted in a singular fit and was removed. The linear model has no random intercept for year or MODIS scene because these produced singular fits. df–degrees of freedom; Δ AIC–difference in AIC to the top model.

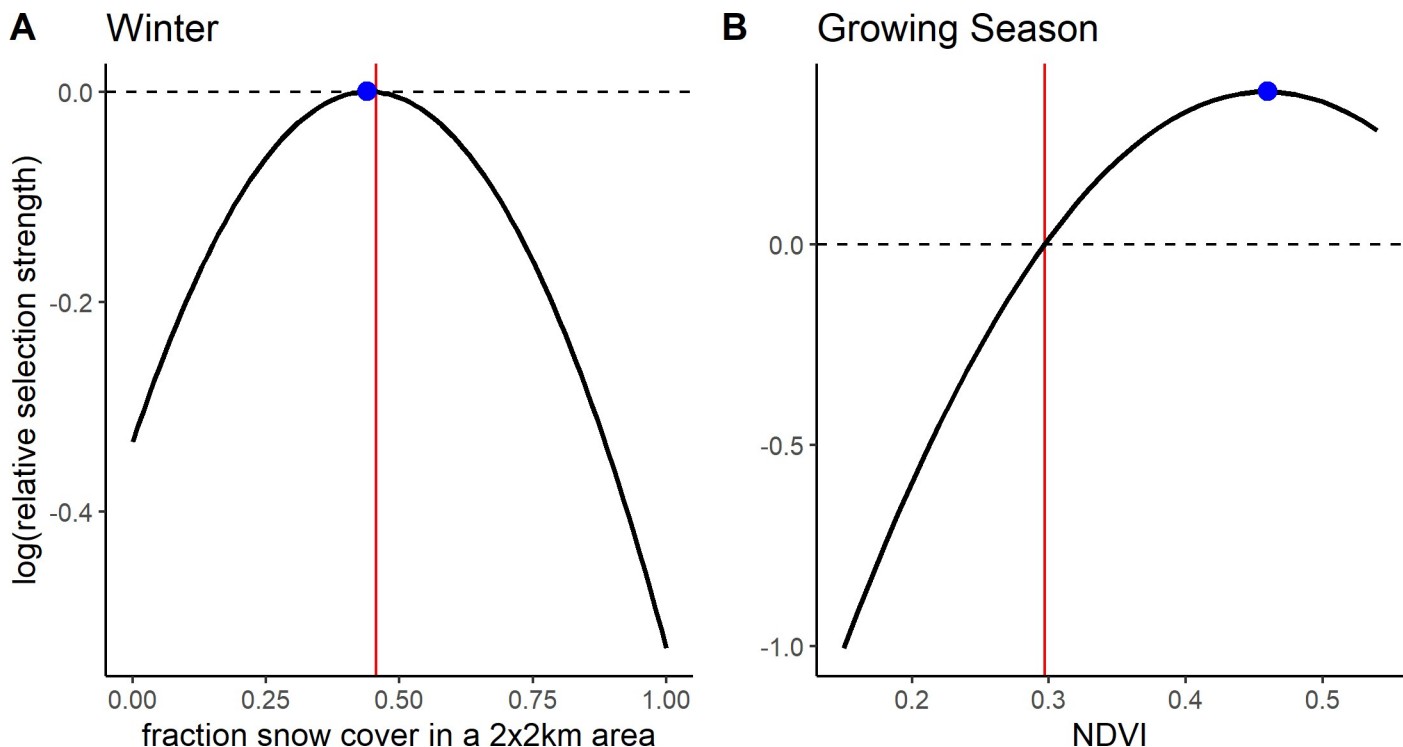

**Fig 2.** Selection by Mongolian gazelles for A) snow cover during winter and B) NDVI during the growing season. RSFs show Mongolian gazelles select for intermediate snow cover in winter (Aug. 30 –May 8) (A). During the growing season (May 8 –Aug. 29), gazelles select for above average NDVI (B). Selection is quantified as relative selection strength (RSS). Curves were plotted based on the best fit RSF for the population as determined by AIC. The vertical line indicates the mean NDVI (fraction snow cover), which was used as the reference point to create the curves. Dots indicate the NDVI (fraction snow cover) value at which RSS peaks. The curve is only plotted for the range of NDVI (fraction snow cover) observed at the used and available points over the study period (Oct. 2014—May 2018), where the upper and lower 5% of values have been removed. Means are also based on this subset.

observed (-0.16 and 0.79 respectively) the selected NDVI value would be more intermediate, although still above the mean which remains 0.30 for the entire data set. During winter we found significant selection for intermediate snow cover values at the population level (Fig 2; $\beta_{\text{snow}}$ = 1.51, $p < 0.001$; $\beta_{\text{snow}}^2$ = -1.70, $p < 0.001$; S3 Table). While our focus was on the population level result (i.e. the average individual), the random effects explained considerable variation, mostly through the random slopes for individual (S4 Table; marginal $R^2$ = 0.003, conditional $R^2$ = 0.41). The selected snow cover value (0.44) was just below the mean snow cover within the population range over the four winters (0.46).

In contrast to the population level analyses, the individual SSFs were not informative. We discuss the results for models with 400 random points because beta coefficients no longer changed much with this amount of random points (S1 Appendix). While p-values still changed, this was only for the non-significant models. Significant models generally showed stable p-values across the range of pseudo-absence points (S1 Appendix). Few individuals showed significant selection at any scale (Fig 3). If sample size were a problem, we would expect to find many more individuals selecting at the 1 day scale then at the 5 and 10 day scale, but this was not the case (see S5–S7 Tables for sample sizes). For the growing season, we did not have sufficient data to run individual-year models for the 5 and 10 day scales, so we report on results from models run by individual at all scales (but see S3 and S4 Figs and S8 Table for the 1 day SSFs run by individual-year). For those individuals who did show selection during the growing season the results were similar to the RSFs, with individuals selecting for above

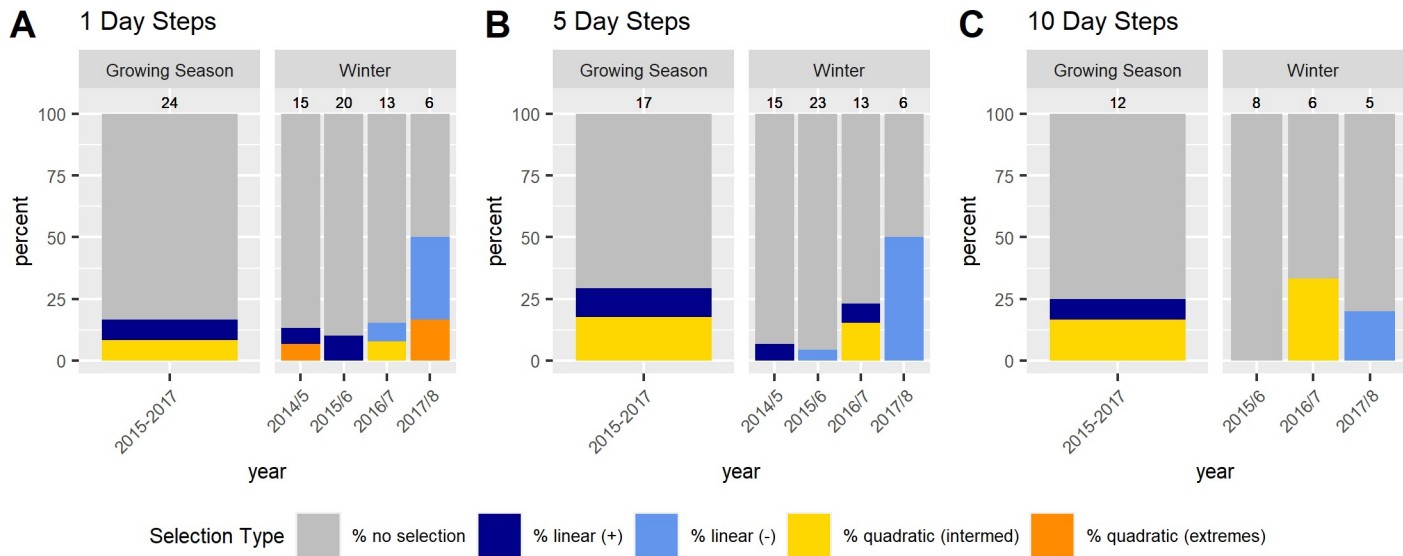

**Fig 3.** Differences among Mongolian gazelles in selection for NDVI during the growing season and for snow cover in winter at different scales (A– 1 day, B– 5 day, C– 10 day step scale) based on individual SSFs. Blue, yellow, and orange sections of the bar plots show the percent of individuals selecting for NDVI during the growing season or for snow cover during winter, grey sections shows the percent of individuals that did not show selection. For those individuals showing selection, dark blue indicates selection for the highest NDVI (or snow cover) available, light blue indicates selection for the lowest NDVI (or snow cover) available, yellow selection for intermediate values, and orange selection for both high and low NDVI (or snow cover). Above each bar are the number of individuals which had sufficient data (minimum of 20 steps) to run step-selection functions. The data show no evidence for selection for the majority of individuals and this trend holds across all scales tested.

average NDVI, often at the upper end of available NDVI, at all scales (Fig 4; S5 and S6 Figs). In the SSFs, this was the result of selection taking a quadratic or linear shape, whereas selection took only a quadratic shape in the RSF (Fig 2 vs. Fig 4; S8 Table). In winter, we had enough data to look at selection by individuals within a year. Some gazelles selected for the maximum snow cover available to them within a year, others minimum snow cover, and others intermediate snow cover (Fig 5; S7 and S8 Figs and S9 Table). This meant that there was selection for snow cover both above and below the mean snow cover during a given year. This mix of selection types and a lack of consistency were seen at all scales (S7 and S8 Figs). We also examined if gazelles that had more than one year of data showed the same selection across years in winter. While most gazelles showed consistent selection in winter (e.g. a positive, linear model was the top model and significant for each year of data), up to a quarter did switch selection strategies (1 day scale = 6 individuals out of 16 with more than one year of data, 5 day scale = 7 out of 17, 10 day scale = 1 out of 4, Fig 6). Switching selection includes switching the shape of selection (e.g., linear to quadratic), the direction of selection (e.g., positive, linear to negative linear), or between selecting and not selecting.

## Discussion

### Selection at the population level

We explored resource selection of nomadic Mongolian gazelles across years and seasons at the population scale and at the individual, daily to weekly scale. During the growing season, at the population level, selection took a quadratic shape with the average gazelle selecting for above average NDVI. We therefore found only limited support for our hypothesis that gazelles select for intermediate NDVI according to the FMH. Here we emphasize that while the selected value of NDVI is an intermediate value of NDVI, which ranges from -0.2 to 1, it is above the

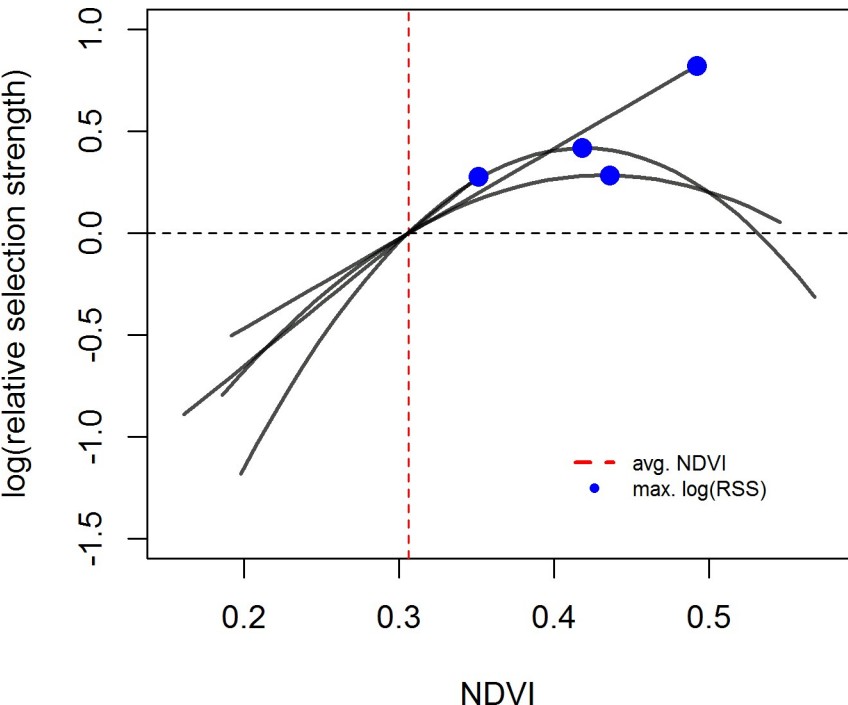

**Fig 4. Selection for NDVI at the 1 day step scale shown by individual Mongolian gazelles during the growing season (May 9 –Aug. 29).** Selection is quantified as relative selection strength (RSS). Curves were plotted based on the results of the individual SSFs and show the best fit model for an individual. Only those individuals who showed significant selection at the 1 day step-scale are shown. The vertical line indicates the mean NDVI (as described in the methods), which was used as the reference point to create the curves. Dots indicate the NDVI values at which RSS peaks. The curve is only plotted for the range of NDVI available to the individual gazelle as defined in the methods. Positive log(RSS) values indicate selection for that NDVI value over the mean NDVI value, negative log(RSS) values indicate avoidance of that NDVI value when compared to mean NDVI. Results for the 5 and 10 day scales can be found in the (S5 and S6 Figs).

mean available NDVI and in the upper range of the available NDVI. NDVI describes vegetation greenness and the observed values in the steppe were generally less than 0.5. Lower NDVI values can reflect areas with more visible bare soil [48, 78, 79]. Such areas partially occur naturally in the steppe, but are also due to drought or overgrazing by livestock [52, 80]. Therefore gazelles might be choosing areas with higher NDVI because these have denser vegetation cover and thus more forage. Therefore with NDVI we might not be able to detect selection for forage quality according to the FMH, but the observed selection suggests that gazelles find areas with higher forage cover. Yet selection decreases again at the upper end of NDVI values, likely because these areas reflect tall, dense vegetation that is difficult to digest and serves as a hiding place for predators such as wolves. While we found only limited support for our original hypothesis in the growing season, we did find support for our hypothesis that gazelles select for intermediate snow cover during winter. This suggests that in winter gazelles avoid areas of high snowfall, but seek out areas that provide enough snow for hydration without restricting movement or access to forage.

At the population level, previous work on Mongolian gazelles found selection for intermediate NDVI in spring and fall [25]. In winter, two separate studies both found avoidance of areas with snow, especially those with a long snow-cover-duration [28, 34]. Discrepancies between these studies and ours could be methodological in origin. [25] and [34] used presence/absence data from line transect surveys to calculate RSFs, while [28] used GPS data from

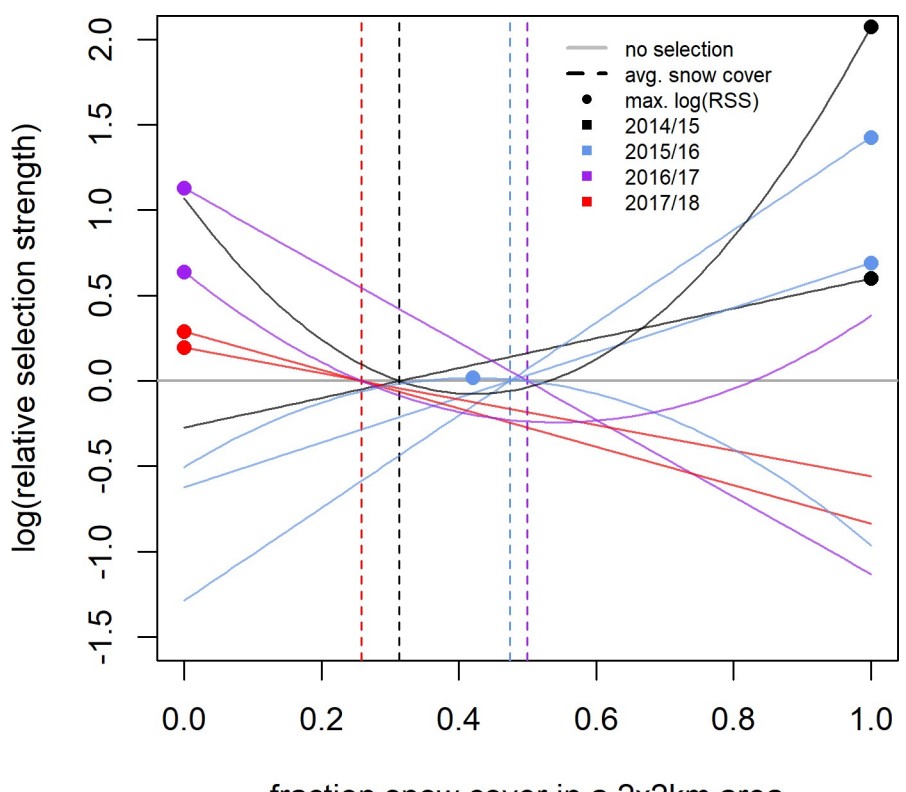

**Fig 5. Selection for snow cover at the 1 day step scale shown by individual Mongolian gazelles in a given year during winter (Aug. 30 –May 8).** Selection was quantified as relative selection strength (RSS). Curves were plotted based on the results of the individual SSFs and show the best fit model for an individual. Only those individuals who showed significant selection at the 1 day step-scale are shown. The vertical lines indicate mean snow cover for a specific year (as described in the methods) and were used as the reference points to create the curves. Dots indicate the snow cover values at which RSS peaks. The curve is only plotted for the range of snow cover available to the individual gazelle as defined in the methods. Positive log(RSS) values indicate selection for that snow cover value over the mean snow cover value, negative log(RSS) values indicate avoidance of that snow cover value when compared to the mean snow cover value. Results for the 5 and 10 day scales can be found in the (S7 and S8 Figs).

8 Mongolian gazelles to build a Maxent habitat suitability model. Despite their differences these studies and ours all indicate that nomadic animals can adjust to unpredictable vegetation during the growing season and find vegetation free from too much snow cover in winter.

## Comparison of population and individual level results

Whereas previous studies have focused on selection for forage at the population level, we also examined selection at the individual level [35–37]. The motivation for this individual level analysis was that Mongolian gazelles captured at the same place and time often exhibit distinct movement paths that take them to very different areas of the steppe. We were interested if individuals nonetheless end up in areas with similar vegetation or snow cover or if individuals show differences that might then be an explanation for these unique movement paths. In other studies that ran both RSFs and SSFs the focus was to examine changes in selection at different scales (e.g., selection of the home range vs. selection within the home range) [81, 82]. Our hypothesis, though, was that selection for vegetation quality and snow cover should be similar at both scales (i.e., within the population seasonal range and along the individual movement path) and among individuals despite their different movement paths. This is because unlike

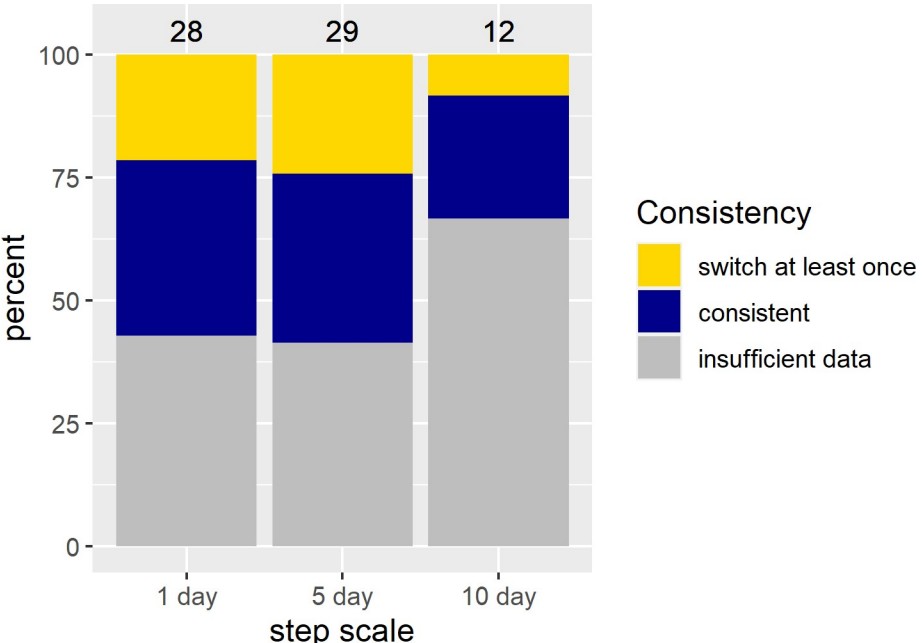

**Fig 6. Percent of Mongolian gazelles that showed different selection strategies for snow cover in different years according to SSFs.** Not all Mongolian gazelles show consistent selection strategies for snow cover across years. For all individual gazelles for which we had more than one year of data during the winter season the percent that switched their selection strategy at least once was calculated. This could be a switch from one type of selection to another (e.g. high snow cover vs. intermediate snow cover) or from selecting to not showing selection. We did not have sufficient data to run individual-year models at all scales during the growing season.

other landscapes that are composed of numerous land cover classes, our landscape simply comprises one habitat class: grassland. While a deer may choose forest for its home range and open meadows within the forest to forage in, such heterogeneity of habitat classes does not exist in the Mongolian steppe. What varies is the quality of the grasses and forbes due to growing state or plant composition and the depth of the snow. Currently there is no readily available remote sensing data to examine these aspects of the steppe. Therefore our interest in testing our hypothesis at the population and individual scale was to account for the individualistic movement paths which are more defined in the SSFs than the RSFs. While we found patterns in selection at the population level and can describe the selection of an average gazelle, our results from the SSFs examining individual level selection were inconclusive. For most individuals in both seasons we could not detect selection. During the growing season, the few individuals for which we were able to detect selection showed a selection for NDVI above the mean, across all scales, which is consistent with the results of the RSFs. The main difference is that selection took both a linear and a quadratic shape. Therefore some individuals showed selection according to the FMH, selecting NDVI in the intermediate range of what was available to them, although NDVI at peak selection was still above the mean of available NDVI. In winter, selection among individuals was variable with some selecting for high, some low, and some intermediate snow cover. Therefore we observed selection for both above and below average available snow cover. This variability was seen at all scales. Individuals that selected for high snow cover might have gotten stuck in deep snow, or snow depth was too low to impact movement and forage. Our inability to detect selection in the majority of individuals was surprising. If sample size (see S5–S7 Tables) were a problem we would have expected to see many more individuals showing selection at the 1 day scale than at the larger scales. While there was

a decrease, it was minimal since so few individuals already showed selection at the 1 day scale (Fig 3). There are several possible explanations, both methodological and biological, which could explain these results. These explanations are discussed below and may shed light on aspects of SSFs that have received little attention yet and should be considered in future research.

### The role of landscape configuration and random searches

One explanation for our inability to detect selection at the individual level using SSFs may be the homogeneity of the landscape across very broad scales. NDVI and snow cover in the steppe change only across broad scales which may lead to only small differences in NDVI or snow cover between gazelle locations and their surroundings within a daily or weekly movement scale (S2 Appendix) [3]. SSFs reduce available resources to those within the selected step scale, which users should carefully choose [74], but which is often determined by the frequency of the GPS fixes. While gazelles cover large distances at the 5 and 10 day scales (27-35km), NDVI variability at these distances may have still been too small to show any significant differences in NDVI between selected and random locations (S2 Appendix). If NDVI at selected and random points are similar, then SSFs would interpret this as no selection, even if the gazelle is in an area with high quality forage. This implies that while Mongolian gazelles may make decisions at sub-daily time scales on which plants to eat based on their digestibility, when it comes to finding the larger area within which they should forage they need to move far before they are in an area with different characteristics. This raises the question of what search strategies can be used when a landscape has large, homogeneous patches. If gazelles were to use perception, this would require perception of resources beyond their sight. The Eastern Steppe is a flat, treeless landscape. In similar landscapes in the Serengeti wildebeest (*Connochaetes taurinus*) were observed moving toward a thunderstorm they heard 24 kilometers away or toward rainstorms that darkened the sky as far as 80 kilometers away [83]. In addition, Mongolian gazelles which have found good resources could potentially call to other gazelles several kilometers away to share that information [84]. Therefore gazelles might be able to find good resource patches, even if they are far away.

Alternatively, given the large patch sizes of vegetation and their unpredictable dynamics using perception might not, or only partially, be feasible, and gazelles may in many cases use random search strategies to find new resources. This is because in unpredictable resource landscapes, memory and gradient following are not expected to be useful in finding resources. Instead, animals might employ area restricted searches [85, 86] that may be less efficient than perception and which might often place Mongolian gazelles in unfavorable habitat and thus make it difficult to detect selection in general.

Whether gazelles use random searches or use perception to choose between large, homogeneous habitat patches–selection for foraging patches is likely to occur only rarely. If this is the case then SSFs might be too fine-scaled to pick up on these rare events. Step-lengths can only be increased to a certain point before sample size becomes limiting or the area examined becomes similar to an RSF approach. Using SSFs with nomadic animals may therefore require using integrated step-selection functions to associate changes in movement (e.g., speed, turn angle) with differences in resource quality [87]. Although, we would still face the challenge of finding a step-length that is appropriate for both the behavior of the animal and the composition of the landscape. For both SSFs and RSFs another useful approach would be to categorize movement trajectories into foraging and searching behavior which would allow for a more focused examination of selection during certain behavioral modes [81, 88, 89]. This might also help us identify over what scales gazelles forage and search for forage. SSFs could then be used

within foraging patches to look at vegetation preferences, but this would require sub-daily movement data and better data on vegetation quality. Yet these categorization methods have not been tested on nomadic animals. Finally, by simulating random searches of a theoretical animal with a known habitat preference, future work could examine what approach is best for studying resource selection in animals that use random search strategies.

## Selection in dynamic landscapes

An additional factor that may complicate the SSF approach is the possibility that resource configuration in the Mongolian steppe may change over time. By using NDVI and snow cover the spatial configuration of resources changes over space and time. How homogeneous or heterogeneous the landscape is will influence at what scale animals can select for vegetation or snow cover and at what scale SSFs can identify selection. If the configuration of resources in space constantly changes, then SSFs, which can only test one scale at a time, might have difficulty in detecting selection. Other studies across a number of taxa have used NDVI for SSFs (Canada lynx [*Lynx canadensis*]–[81]; pronghorn [*Antilocapra americana*]–[90]; Hartmann's mountain zebra [*Equus zebra hartmannae*]–[91]; African elephants [*Loxodonta africana*]–[92]), but no study indicated similar challenges with NDVI. All found selection for NDVI, although the time steps examined were much smaller than in our study ($< = 4$hrs). Smaller time steps result in more data that might have made it easier to detect selection, given the environment is sufficiently heterogeneous to distinguish selected from available points. For all studies NDVI was only one of many variables tested and we do not have enough information about the dynamics and configuration of the respective landscapes to draw comparisons to our study. Future work could use simulations of landscapes and animal movements to examine how landscape configuration influences SSFs' ability to detect selection.

## Variable selection

For those individuals where SSFs detected significant selection, the direction and degree of selection for NDVI and snow cover was variable and inconsistent. Across years individuals showed different selection patterns. Consistent differences in selection among individuals are not unusual and are often associated with animal personalities [35]. To be classified as a personality, individuals must display the same selection patterns across many years [35]. Because our results did not indicate consistency across years, they are difficult to interpret. In winter, variability is most likely due to data limitations that prevent us from distinguishing snow depth. Individuals that showed selection for high snow cover might have been in areas with shallow snow depth. In both seasons one biological explanation for the variability in selection could be that differences in selection among individuals or differences within individuals across years might be due to an individual's physiological state. For example, reproductive state, sex, age, thermal stress, changing energy requirements, and health have all been shown to affect selection strategies of animals [30, 93–95]. Individual gazelles may therefore have different selection strategies depending on their current state which could cause differences among individuals as well as differences within individuals across years. Thus, in some cases it may be beneficial to maximize energy intake by selecting for intermediate biomass vegetation. In other cases, selecting high biomass vegetation can allow an individual to meet its basic energy requirements quickly, leaving time for other activities such as reproduction [30, 96, 97]. In addition, seasonal and patchy presence of biting insects, human activity, the need to protect calves, droughts, and heat waves could all alter selection strategies for longer time periods.

## Unexamined drivers of selection

Another simple explanation for the inconclusive SSF results might be that other factors are driving selection at a local scale. Wolves, biting flies, human disturbance, and livestock grazing likely all alter how gazelles select for vegetation and/or snow cover. Because the presence of all these factors changes in space and time they are difficult to measure and we were not able to account for them. Yet, [51] showed that although there are fewer Mongolian gazelles in areas of intermediate NDVI with more humans, selection for intermediate NDVI is still detectable. Thus, we would expect additional biotic and abiotic factors to weaken but not eliminate our ability to detect selection for vegetation or snow cover. It might have also been easier to detect selection if data on the digestibility or composition of the vegetation was available for the growing season and data on snow depth was available for winter. NDVI and snow cover are the best available metrics that are available across time and at broad scales, but are only rough proxies for habitat quality. Better proxies for vegetation quality and data on snow depth in combination with temperature data would be particularly valuable to examine responses of Mongolian gazelles to climate change.

## Conclusion

Our conclusions on how nomadic Mongolian gazelles track vegetation resources and snow cover are defined by the population-level RSF. Only at this scale were we able to detect selection for NDVI and snow cover. Despite the unpredictable landscape and unique movement trajectories, the RSFs show Mongolian gazelles do select areas with similar vegetation during the growing season and similar snow cover during winter. Therefore there appears to be similarity in selection among individuals. While we would need data on the digestibility or species composition of the vegetation to understand why gazelles selected for higher than average NDVI values, it is striking that nomadic individuals tracking unpredictable vegetation nonetheless find similar areas of vegetation. Similarly, in winter, the population selected for intermediate snow cover. Interestingly, we were not able to detect selection for most individuals using SSFs. This may bring attention to aspects of SSFs that yet need to be studied and we hope our study inspires future research into the how landscape configuration and search strategy may influence the performance of SSFs which in turn would help to better understand resource selection of herbivores in grassland systems. This would be especially important for nomadic animals where SSFs could help us better understand the unique movement paths that individuals take and if they are related to fine-scale resource selection.

## Supporting information

**S1 Fig. Histogram of 1, 5, and 10 day displacement distances (in km) from 34 Mongolian gazelles in the Eastern Steppe of Mongolia.** Displacement is the straight line distance between two GPS points. It does not account for the distance the individual actually traveled to move between those two points.
(TIFF)

**S2 Fig. The effect of the number of pseudo-absence points on model coefficients of the RSFs.** Models were run with an equal number of presence and pseudo-absence points, double the number of pseudo-absence points as presence points, and 5, 10, and 20 times the number of pseudo-absence points as presence points. This was done for all models. Here we show the results for the RSF for each season which had the most support based on AIC. A) For the growing season this was a quadratic model (NDVI + NDVI$^2$) with a random effect for NDVI scene. B) For winter, this as a quadratic model (snow cover + snow cover$^2$) with a random slope and

intercept for individual gazelles and a random intercept for year. Beta coefficients for the linear and quadratic parameter are shown. Differences between 10 and 20 times the number of pseudo-absence points are small so we present results of models run with 20 times more pseudo-absence points than presence points.
(TIFF)

**S3 Fig. Selection for NDVI at the 1 day step scale shown by individual Mongolian gazelles in a given year during the growing season (May 9 –Aug. 29).** Selection is quantified as relative selection strength (RSS). Curves were plotted based on the results of the individual-year SSFs and show the best fit model for a given individual-year combination. Only those individuals who showed significant selection at the 1 day step-scale are shown. The vertical lines indicate the mean NDVI for a given year (as described in the methods) and were used as the reference points to create the curves. Dots indicate the NDVI values at which RSS peaks. The curve is only plotted for the range of NDVI available to the individual gazelle as defined in the methods. Positive log(RSS) values indicate selection for that NDVI value over the mean NDVI value, negative log(RSS) values indicate avoidance of that NDVI value when compared to the mean NDVI value.
(TIFF)

**S4 Fig. Differences among Mongolian gazelles in selection for NDVI during the growing season at the 1 day step scale based on step-selection functions (SSFs) run for each individual-year combination.** The majority of Mongolian gazelles did not show selection at the 1 day step scale. Blue, yellow, and orange sections of the bar plots show the percent of individuals selecting for NDVI during the growing season of a given year or for snow cover during winter of a given year, grey sections show the percent of individuals that did not show selection. For those individuals showing selection, dark blue indicates selection for the highest NDVI (or snow cover) available, light blue indicates selection for the lowest NDVI (or snow cover) available, yellow selection for intermediate values, and orange selection for high and low values. Above each bar are the number of individuals which had sufficient data (minimum of 20 steps) to run step-selection functions on an individual-year basis.
(TIFF)

**S5 Fig.** Selection for NDVI at the 5 day step scale shown by individual Mongolian gazelles during the growing season (May 9 –Aug. 29). Selection is quantified as relative selection strength (RSS). Curves were plotted based on the results of the individual SSFs and show the best fit model for an individual. Only those individuals who showed significant selection at the 5 day step-scale are shown. The vertical line indicates the mean NDVI (as described in the methods), which was used as the reference point to create the curves. Dots indicate the NDVI values at which RSS peaks. The curve is only plotted for the range of NDVI available to the individual gazelle as defined in the methods. Positive log(RSS) values indicate selection for that NDVI value over the mean NDVI value, negative log(RSS) values indicate avoidance of that NDVI value when compared to the mean NDVI value.
(TIFF)

**S6 Fig. Selection for NDVI at the 10 day step scale shown by individual Mongolian gazelles during the growing season (May 9 –Aug. 29).** Selection is quantified as relative selection strength (RSS). Curves were plotted based on the results of the individual SSFs and show the best fit model for an individual. Only those individuals who showed significant selection at the 10 day step-scale are shown. The vertical line indicates the mean NDVI (as described in the methods), which was used as the reference point to create the curves. Dots indicate the NDVI values at which RSS peaks. The curve is only plotted for the range of NDVI available to the

individual gazelle as defined in the methods. Positive log(RSS) values indicate selection for that NDVI value over the mean NDVI value, negative log(RSS) values indicate avoidance of that NDVI value when compared to the mean NDVI value.
(TIFF)

**S7 Fig. Selection for snow cover at the 5 day step scale shown by individual Mongolian gazelles in a given year during the winter (Aug. 30 –May 8).** Selection was quantified as relative selection strength (RSS). Curves were plotted based on the results of the individual SSFs and show the best fit model for an individual. Only those individuals who showed significant selection at the 5 day step-scale are shown. The vertical lines indicate mean snow cover in a given year (as described in the methods) and were used as the reference point to create the curves. Dots indicate the snow cover values at which RSS peaks. The curve is only plotted for the range of snow cover available to the individual gazelle as defined in the methods. Positive log(RSS) values indicate selection for that snow cover value over the mean snow cover value, negative log(RSS) values indicate avoidance of that snow cover value when compared to the mean snow cover value.
(TIFF)

**S8 Fig. Selection for snow cover at the 10 day step scale shown by individual Mongolian gazelles in a given year during the winter (Aug. 30 –May 8).** Selection was quantified as relative selection strength (RSS). Curves were plotted based on the results of the individual SSFs and show the best fit model for an individual. Only those individuals who showed significant selection at the 10 day step-scale are shown. The vertical line indicates mean snow cover (as described in the methods), which was used as the reference point to create the curves. Dots indicate the snow cover values at which RSS peaks. The curve is only plotted for the range of snow cover available to the individual gazelle as defined in the methods. Positive log(RSS) values indicate selection for that NDVI value compared to the mean, negative log(RSS) values indicate avoidance of that NDVI valued compared to the mean.
(TIFF)

**S1 Table. A summary of the data available for each individual Mongolian gazelle.** Information is provided for each Mongolian gazelle on sex, the number of daily points recorded, the day we began tracking the individual and when we stopped, the duration of the data in years and months, and the frequency at which GPS locations were recorded.
(XLSX)

**S2 Table. Beta coefficients for the best fit RSF model describing selection by Mongolian gazelles for NDVI during the growing season (May 9 –Aug. 29).** The quadratic model was based on 3379 daily locations from 25 individuals over 3 years. The table presents the beta coefficients, standard error, and p-values for the overall model.
(XLSX)

**S3 Table. Beta coefficients for the best fit RSF model describing selection by Mongolian gazelles for snow cover during winter (Aug. 30 –May 8).** The quadratic model was based on 9791 daily locations from 33 individuals over four winters. The table presents the beta coefficients, standard error, and p-values for the overall model.
(XLSX)

**S4 Table. Variance (with standard deviation) explained by the random effects in the RSF model describing selection by Mongolian gazelles for snow cover during winter (Aug. 30 –May 8).**
(XLSX)

**S5 Table. Number of steps for the growing season SSFs for each individual gazelle over the three step-length scales tests.**
(XLSX)

**S6 Table. Number of steps for the growing season SSFs run by individual/year at the 1 day step-length scale.**
(XLSX)

**S7 Table. Number of steps for the winter SSFs run by individual/year over the three step-length scales tests.**
(XLSX)

**S8 Table. AIC values for the different growing season (May 9 –Aug. 29) SSF models examining how Mongolian gazelles select for NDVI.** Models represent different hypotheses on how Mongolian gazelles select for NDVI (linear or quadratic). A linear model has just NDVI as an explanatory variable. A quadratic model has NDVI + NDVI$^2$ as explanatory variables. Models were run for each individual with 1, 5, and 10 day step-lengths. For the 1 day step length we were also able to run models by individual-year. Results are shown only for those individuals where the top-model had a significant p-value ($\alpha < 0.05$). $\Delta$AIC is the difference between a model and the top model.
(XLSX)

**S9 Table. AIC values for the different winter (Aug. 30 –May 8) SSF models examining how Mongolian gazelles select for snow cover.** Models represent different hypotheses on how Mongolian gazelles select for snow cover (linear or quadratic). Models were run for each individual-year with 1, 5, and 10 day step-lengths. Results are shown only for those individual-years where the top-model had a significant p-value ($\alpha < 0.05$). $\Delta$AIC is the difference between a model and the top model.
(XLSX)

**S1 Appendix. Effect of the number of random points on step-selection function coefficients.**
(PDF)

**S2 Appendix. Analysis examining differences in NDVI between areas selected by and available to Mongolian gazelles.**
(PDF)

## Acknowledgments

We would like to thank Jörg Albrecht and Johannes Signer for their statistical insight and assistance. It would not have been possible to collect the data on Mongolian gazelles without the help of numerous field assistants, the World Wide Fund for Nature (WWF Mongolia), which helped to obtain the animal capture permits, and the Ministry of Environment and Tourism of Mongolia.

## Author Contributions

**Conceptualization:** Theresa S. M. Stratmann, Thomas Mueller.

**Data curation:** Theresa S. M. Stratmann, Nandintsetseg Dejid.

**Formal analysis:** Theresa S. M. Stratmann.

**Funding acquisition:** Theresa S. M. Stratmann, Nandintsetseg Dejid, Kirk A. Olson, Thomas Mueller.

**Investigation:** Theresa S. M. Stratmann, Nandintsetseg Dejid, Kirk A. Olson, Thomas Mueller.

**Methodology:** Theresa S. M. Stratmann, Nandintsetseg Dejid, Justin M. Calabrese, William F. Fagan, Christen H. Fleming, Kirk A. Olson, Thomas Mueller.

**Project administration:** Theresa S. M. Stratmann, Thomas Mueller.

**Resources:** Thomas Mueller.

**Software:** Theresa S. M. Stratmann.

**Supervision:** Thomas Mueller.

**Validation:** Theresa S. M. Stratmann.

**Visualization:** Theresa S. M. Stratmann.

**Writing – original draft:** Theresa S. M. Stratmann.

**Writing – review & editing:** Theresa S. M. Stratmann, Nandintsetseg Dejid, Justin M. Calabrese, William F. Fagan, Christen H. Fleming, Kirk A. Olson, Thomas Mueller.

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
