## [Decision Letter · Decision Letter 0]

24 Nov 2020

PONE-D-20-30912

Resource selection of a nomadic ungulate in a dynamic landscape

PLOS ONE

Dear Dr. Stratmann,

Thank you for submitting your manuscript to PLOS ONE. After careful consideration, we feel that it has merit but does not fully meet PLOS ONE’s publication criteria as it currently stands. Therefore, we invite you to submit a revised version of the manuscript that addresses the points raised during the review process.

Both reviewers felt the manuscript was well-written, the study design sound, and the article addressed a little studied topic. Reviewer 1 mainly had editing suggestions which reviewer 2 did as well. Reviewer 2, however, had methodological concerns that need to be considered and addressed in your revision or offer clearly stated rebuttals. I have one comment about labeling the species as nomadic. After reading your manuscript, in particular the life history of Mongolian gazelles, I believe that Mongolian gazelles are nomadic. But I wondered why a metric such as net displacement was not calculated and reported to quantify nomadic behavior. I foresee a day when nomadic behavior is more thoroughly studied, and a common metric summarizing nomadic behavior across studies might lead to work that more fully examines the causes and consequences of this behavior. 

We look forward to receiving your revised manuscript.

Kind regards,

Floyd W Weckerly

Academic Editor

PLOS ONE

2. In your Methods section, please provide additional location information of the study area, including geographic coordinates for the data set if available.

Reviewers' comments:

Reviewer's Responses to Questions

**Comments to the Author**

1. Is the manuscript technically sound, and do the data support the conclusions?

Reviewer #1: Yes

Reviewer #2: Yes

2. Has the statistical analysis been performed appropriately and rigorously? 

Reviewer #1: Yes

Reviewer #2: Yes

3. Have the authors made all data underlying the findings in their manuscript fully available?

Reviewer #1: No

Reviewer #2: Yes

4. Is the manuscript presented in an intelligible fashion and written in standard English?

Reviewer #1: Yes

Reviewer #2: Yes

5. Review Comments to the Author

Reviewer #1: Stratmann et al examined habitat selection by Mongolian gazelles as it relates to selection of snow cover in winter and NDVI in summer at several spatial scales using RSFs and SSFs. Overall, I found the manuscript well written and clear. I do however have a few methodological concerns which I outline below, along with several minor comments.

L219-226: I’m a bit confused here. Why were data reclassified to just snow/no-snow? Would it not be more accurate to maintain the snow cover extent data in the original layer instead of seemingly making each pixel binary? And if this was done, was there a cutoff? If a pixel had, say 10% snow cover extent, was this classified as “snow”? If so, this would make for a highly inflated mean snow cover extent at the 5 X 5 km scale, as each pixel could be 10% and the result would be an area with 100% snow cover. 5 X 5 km also seems like a very coarse scale to use here. If this is the average that a gazelle walks in a day, they can really only select for a very limited number of pixels that are in the immediate vicinity of the pixel they are in at the start of the day (and likely wouldn’t have any information about many neighboring pixels).

L246-253: This seems strange to me. By drawing your point outside the daily movement range buffer of the location, you are essentially excluding all the habitat that that gazelle could plausibly select as an alternative resource, while only including spatial data that is likely outside where the individual can select.

L255-256: I don’t believe a normal GLMM (or GLM) is sufficient here. Individual locations correspond with available locations drawn at the same timeframe. That is, points are matched temporally. This data is lost in a typical regression, where an animal can still “choose” different points in time. I think this model should be a conditional logistic regression using either ‘clogit’ as was done for SSFs, or using the method developed by Muff et al (2020), which would permit the addition of random terms.

Minor comments:

L34/35: selection “at” the population/individual level?

L36: Why “spatio-“ here? This line only references differences in temporal scales.

L67: Why “interact to”? Are plant characteristics interacting, or is it simply that plant characteristics influence herbivore selection?

L77: “To work for” isn’t very clear. I suggest recasting this sentence, or simply cutting the first five words of the sentence.

L90, 126, 206: “Mongolian gazelles”

L113/114: References?

L177: I suggest changing this sentence a bit. This wording could mean that there are no open plains or rolling hills.

L187: Individuals cannot adapt, as this is a result of natural selection across generations. To make this sentence clearer, I suggest either using the term “acclimate”, or changing this to something along the lines of “gazelles must therefore be adapted to cope with a wide range of conditions”, depending on which meaning you wish to portray.

L190-191: NDVI is a value, not snow cover. How was snow cover determined from the NDVI layers? I assume based on the snow “flag”/layer of MODIS NDVI, but I think this should be explained. To be clear, this was when no pixels in the entire study area had a snow cover flag?

L209: Should this be “rarefied” instead of summarized?

L236-237: Why did you crop MCPs? Was there a biological reason to exclude points outside Mongolia? What proportion of points did this represent?

L277: “and snow”?

L293: “allowed us to sample”

L304-306: This is confusing. This describes the procedure for the winter models, yet the example is from a gazelle in spring? Obviously these are related, but should this be temporal scale instead of spatial (the spatial scale is changing because the temporal one is).

L308: Typo in “therefore”.

L348: “At” the population level?

Table 1: The AIC Weight for growing season doesn’t make sense. For the top model to be 0.66 and the 2nd highest to be <0.001, then there has to be a lot of other models with small (but not insignificant) weights to make up the difference, Unless these are not the two top models?

L391: I don’t follow this. Weren’t linear and quadratic the only shapes tested? So isn’t this statement guaranteed to be the case?

L392-395: I think this needs to be reworked. I’m guessing the first sentence is suggesting a variety of responses across individuals, with some exhibiting positive and some exhibiting negative linear selection, and some with the downward parabola. This should be more explicitly explained. Stating that gazelles selected for max, min, and intermediate levels sounds like they simply select everything (see also lines 495-497). Is the 2nd sentence true for individuals who selected an intermediate amount of snow cover?

L452: Remove first “for” in “to detect for selection for forage”.

L454, 610: Change “preference” to selection. Preference typically is used to describe a situation where selection is quantified when availability is equal (e.g., a cafeteria trial).

L457-459: This is a very short disjointed paragraph with really no discussion per se.

L460: This seems early in the Discussion to be conjecturing about differences between your study and others. I suggest moving this to near the end of the discussion.

L469: Citation?

L574: What is meant by “important”? Important for what? For these to be classified as animal personalities?

Fig. 2: The overall resolution of this figure is quite low. The two shades of blue are also quite difficult to tell apart, especially in bars with only a single color.

Fig. 4: Define the horizontal dashed line in the caption and legend and make it a different line type from the vertical line for snow cover in 2014/15.

Well done, this was an enjoyable paper to read.

Reviewer #2: The manuscript “Resource selection of a nomadic ungulate in a dynamic landscape” addresses interesting topic of evaluating resource selection of nomadic Mongolian gazelle. The study provides an important contribution to this topic by addressing selection at multiple spatio-temporal scales using a solid sample size of observational data and appropriate modelling techniques. The strengths of the paper are the strong conceptual framework used to understand nomadic spatial behaviour in ungulates, which is currently understudied. And the complex evaluation of such behaviour across multiple spatial and temporal scales will result in this paper being important. The weakness, which I’m sure you are aware, is the use of proxies, such as NDVI for forage, and snow cover for snow depth. However, this is overshadowed to me by the importance and the rarity of studies on habitat selection of nomadic ungulates in particular, and the methods you used which are appropriate and rigorous. Finally, I think a short statement in the discussion regarding possible changes in behavior of nomadic Mongolian gazelle with climate change could help to improve the visibility of the paper. The study is generally well-perceived, concisely written and well organized. Thus, my concerns are relatively minor and mostly address unclear passages in the text.

Altogether I recommend only minor revisions of the manuscript.

Bellow, I list my concerns in more detail (I also included my comments as the “sticky notes” directly into the pdf of the manuscript and highlighted the related text with a yellow background):

L 56 – “That which is known” - consider re-phrasing

L 56 – “seasons” instead of “season”?

L 64 – … we used Mongolian gazelles… - use either "as a model species" or "for this case study"...

L 77 – 79 – this is a bit confusing... FMH works for both, residents and migrants... both can select for intermediate forage biomass during whole growing season... try re-phrasing

L 98 – unclear... it seems like you mean "trade-offs in selection between habitat types"

L 107 – 108 – I suggest improving, resp. re-phrasing this sentence

L 114 – 115 – ..."to conclude"... instead of …”to arrive at an overall conclusion”…

L 126 – 130 – I think this description is more suitable for Methods/Study area description

L 140 - …”in a fission-fussion manner”… - Im not sure what this term supposed to mean within this context.

L 162 – 163 – sounds like Methods

L 186 – 196 – I dont think this should go to Study area. This is the setting of the temporal scale of your study, it should be under description of the RSFs.

L 207 – 208 – How long were animals tracked? min - max months, average +-SD... How many points were collected from 1 animal? min - max, average +-SD

L 222 – This sentence would need a citation.

L 233 – ..."we generated"... instead of …”we had to generate”…

L 245 – I suggest re-phrasing the beginning of the sentence...

L 292 – missing space - ..."and 10 days..."

L 293 – ..."allowed us"... instead of …”lets us”…

L 322 – 326 – Id say this is already Results...

L 398 – By switching selection strategies, do you mean switching between linear and quadratic selection? And also, do you mean selection for NDVI or snow?

L 454 – ... selection for higher NDVI... instead of ... preference for higher NDVI...

6. PLOS authors have the option to publish the peer review history of their article (what does this mean?). If published, this will include your full peer review and any attached files.

Reviewer #1: No

Reviewer #2: No

---

## [Author Response · Author response to Decision Letter 0]

22 Jan 2021

PONE-D-20-30912

Resource selection of a nomadic ungulate in a dynamic landscape

PLOS ONE

Dear Dr. Weckerly, 

Please find enclosed a revised version of our manuscript PONE-D-20-30912, “Resource selection of a nomadic ungulate in a dynamic landscape.”

We thank you and both reviewers for your positive and valuable comments on the previous version of the manuscript and the opportunity to submit a revised version. The reviewers highlighted the importance of the topic and the rigor of our analyses while also suggesting a number of modifications. In response to your and the reviewers’ comments we have rewritten sections of the manuscript that pertain to nomadism, changed wording of sentences and order of paragraphs to improve clarity, added a figure of our study area, created supplementary material to describe the data better, and have modified several analyses. In particular, the modifications for the analyses involved: changing the spatial resolution of the snow cover data, adding a random effect for NDVI in the RSFs (although this only works for one season), and eliminating the buffering of presence points when sampling absence points for the RSFs. None of these analytical changes have qualitatively changed the results and thus our discussion and conclusions remain the same.

Please find our point by point responses to your and the reviewers’ comments below. We have worked hard in hopes that our revisions effectively respond to the your and the reviewers’ concerns. 

Thank you very much for your consideration.

Sincerely,

Theresa Stratmann (for all authors)

 

Below are our detailed responses to the suggestions we received. The original comment is in italics and our response follows in normal text. Line numbers we cite refer to line numbers in the document *without* track changes. 

Response to Comments from the Academic Editor:

Both reviewers felt the manuscript was well-written, the study design sound, and the article addressed a little studied topic. Reviewer 1 mainly had editing suggestions which reviewer 2 did as well. Reviewer 2, however, had methodological concerns that need to be considered and addressed in your revision or offer clearly stated rebuttals. I have one comment about labeling the species as nomadic. After reading your manuscript, in particular the life history of Mongolian gazelles, I believe that Mongolian gazelles are nomadic. But I wondered why a metric such as net displacement was not calculated and reported to quantify nomadic behavior. I foresee a day when nomadic behavior is more thoroughly studied, and a common metric summarizing nomadic behavior across studies might lead to work that more fully examines the causes and consequences of this behavior. 

We fully agree that using a standardized way to quantify nomadic behavior is important and there is a growing body of literature examining the best methods to achieve this. Past studies have quantified the nomadic behavior of Mongolian gazelles using a number of metrics (e.g., range size, seasonal shifts in range use, pairwise distance between points, variograms to categorize movement, revisits) that can be calculated for any species (Olson et al. 2010, Mueller et al. 2011, Fleming et al. 2014, Nandintsetseg et al. 2019a,b). None of these papers chose to use net squared displacement (NSD), because NSD is problematic for several reasons. In their paper, Fleming and co-authors (2014) show (in the supplementary material section A.1) that their variogram approach is superior to a metric like NSD because it can harness more of the data. Other papers have pointed out that the NSD method has proven difficult to apply to real data sets (Bastille-Rousseau et al. 2016, Cagnacci et al. 2016) and has weaknesses that must be carefully addressed (Bastille-Rousseau et al. 2016, Singh et al. 2016). We also see a conceptual problem with NSD for nomadic species that moves extremely long distances. While our gazelles do not use areas in a repeatable manner and use different and often novel areas each year (Olson et al. 2010, Nandintsetseg et al. 2019a,b), their range does not expand forever as indicated in Fig. 1 of Bunnefeld et al. (2011) because 1) Mongolia is fenced and gazelles cannot move to Russia or China and 2) at some point the habitat becomes unsuitable, bounding the area which gazelles can move to. Until more studies are done on standardized metrics to quantify nomadism, the analyses done in the papers above best quantify the nomadic behavior of the gazelles studied in this project. These analyses can be done for any other species as comparison, in fact Mueller et al. 2011 and Nandintsetseg et al. 2019b compare Mongolian gazelles with other ungulates by measuring the same metrics for all species. 

As we agree that it is important quantify nomadic behavior, we changed the text to emphasize the body of literature that has quantified the nomadic behavior of gazelles and compared it to other species as follows (lines 138-144):

“Mongolian gazelles are considered nomadic based on previous quantitative analyses of GPS location data [3, 37, 42-44]. [43] showed that it takes a gazelle several months to cross the area it uses within the steppe, which can be up to three-times as large as the Serengeti-Mara ecosystem. It has also been shown that individual Mongolian gazelles do not return to specific locations, not even during calving time or winter [37, 42, 44]. The large area used by gazelles, the long time it takes to cross that area, and the lack of repeatability in their movement are the key characteristics of their nomadic movement.”

In your Methods section, please provide additional location information of the study area, including geographic coordinates for the data set if available.

Since our study area is large and thus difficult to represent with geographic coordinates we added a map to better show the study area and the gazelle locations within the study area (Figure 1, lines 209-210). This figure also clarifies a later comment why we limited the available area in the RSFs to be within Mongolia. 

Data availability

We have clarified our data availability statement: 

“For conservation concerns it is not standard to publish the raw GPS data on animal locations, but we provide a data frame with re-centered GPS points, all absence points, and all associated environmental data. This data set is sufficient to run the statistical analyses and reproduce the figures. The data will be made available on Open Science Framework when the manuscript has been accepted and we will provide the DOI for the uploaded data.”

Response to Reviewer #1: 

Stratmann et al examined habitat selection by Mongolian gazelles as it relates to selection of snow cover in winter and NDVI in summer at several spatial scales using RSFs and SSFs. Overall, I found the manuscript well written and clear. I do however have a few methodological concerns which I outline below, along with several minor comments.

We were glad that you enjoyed the paper and feel that your suggestions, which resulted in several changes to the methodology, have improved the quality of manuscript. 

L219-226: I’m a bit confused here. Why were data reclassified to just snow/no-snow? Would it not be more accurate to maintain the snow cover extent data in the original layer instead of seemingly making each pixel binary? And if this was done, was there a cutoff? If a pixel had, say 10% snow cover extent, was this classified as “snow”? If so, this would make for a highly inflated mean snow cover extent at the 5 X 5 km scale, as each pixel could be 10% and the result would be an area with 100% snow cover. 

We appreciated the comment because we now realize that our wording and the name of the data set were confusing: The raw MOD10A2 data set is indeed a categorical raster layer where each 500m x 500m pixel can have one of several values: snow, no snow, missing data, no decision, night, lake, ocean, cloud, lake ice, detector saturated, or fill. No data on percent or extent of snow cover within a pixel is provided. The original data set is essentially binary, with some error values and we emphasize that we did not convert a continuous measure to a binary one. In addition the value for each pixel is based on observations over 8 days. If the pixel had snow in it on one day it is marked as snow. This is another weakness of the data set, but is currently the best way to deal with data gaps due to cloud cover. We changed the text in the main manuscript to avoid any misunderstanding (lines 222-228).

5 X 5 km also seems like a very coarse scale to use here. If this is the average that a gazelle walks in a day, they can really only select for a very limited number of pixels that are in the immediate vicinity of the pixel they are in at the start of the day (and likely wouldn’t have any information about many neighboring pixels).

We agree with the reviewer that the scale could be problematic. To address this problem we re-classified the snow data set to a 2 x 2km resolution so that more pixels could be within a daily step (see lines 228-229). Although the details of the individual SSFs changed, it did not change our overall conclusion. The conclusions for the RSFs also did not change (see figures in next response). Nonetheless we will keep the finer resolution based on these valid concerns. We updated all figures, tables, and supplementary material accordingly. 

L246-253: This seems strange to me. By drawing your point outside the daily movement range buffer of the location, you are essentially excluding all the habitat that that gazelle could plausibly select as an alternative resource, while only including spatial data that is likely outside where the individual can select.

Since the RSF addresses selection at the population scale and we used one point to represent a daily location, our original rationale was to buffer the location by the average daily displacement because the gazelle could have been in that area on that day and so this area could include several presence points. Therefore we did not want to draw absence points from this area. In the literature both approaches are present, with the approach of not buffering having perhaps more support (Bartbet-Massin et al. 2013). Based on your comment, we re-ran the analyses without buffering the presence points and found that it does not change our results in a qualitative way. We now include the simpler procedure without buffering the presence points (see changes to lines 263-269).

Below we show how our changes (snow cover in a 2x2km area and no buffering of presence points) change the results of the RSFs. Figure III is now used in the manuscript: 

I. Original figure:

II. Figure using snow data aggregated at 2x2km area, with a random intercept for NDVI during the growing season, and with buffered presence points: 

III. Figure using snow data aggregated at 2x2km area, with a random intercept for NDVI during the growing season, without buffered presence points: 

Below is an example how the SSFs changed: 

I. Original Figure 4 – Selection for snow cover in the winter at the 1 day step scale. 

II. Results with fraction snow cover in a 2x2km area vs. 5x5km area. We still see the variability in selection that we saw originally, with some individuals selecting for high snow cover, some for low, and some for intermediate.

L255-256: I don’t believe a normal GLMM (or GLM) is sufficient here. Individual locations correspond with available locations drawn at the same timeframe. That is, points are matched temporally. This data is lost in a typical regression, where an animal can still “choose” different points in time. I think this model should be a conditional logistic regression using either ‘clogit’ as was done for SSFs, or using the method developed by Muff et al (2020), which would permit the addition of random terms.

The point of our study is to examine selection at two different scales: at the population level – which zooms out and looks at the broader landscape the animal uses – and the individual level – which focuses on the movement path of the individual. To accomplish this we use a RSFs to examine population level selection within the steppe and a SSFs to examine individual level selection along the movement path. Using these different methods to examine different levels of selection is a standard tool in the literature and thus make our study comparable to others (recently: Reinking et al. 2019, Lamont et al. 2019, and Jakes et al. 2020 [which has a nice figure illustrating this concept]). In the RSF we therefore draw pseudo-absence points from within the entire seasonal range of individuals and then look for a trend across all individuals. In the SSF we narrow down the comparison points to places the gazelles could have reached (in 1, 5, or 10 days) but did not go to at that point in time. In other words, what kind of areas do gazelles choose within the Eastern Steppe (RSF) vs. what area does the gazelle choose as it moves at small time scales (1, 5, 10 days). Therefore for the RSF, while we made sure that the absence points for each presence point came from the same MODIS scene (lines 262-269), we do not want to match points in time with a ‘clogit’ because we consider the entire area used by the gazelles as available because we are no longer restricted to time steps, we rather consider where a gazelle was located within a season. Therefore, throughout the manuscript we tried to make this distinction more apparent (e.g., 176-180, 247-248, 295-296, 511). We also tried to improve clarity within the methods section (lines 262-269). In addition, based on your comments we added a random intercept for MODIS scene to our RSFs. This works in the growing season models because one NDVI scene is 16 days and so there is sufficient data within each scene to estimate a random effect. Adding this random intercept does not qualitatively change our results (see figures above). In winter the MODIS scenes are only 8 days and so we have much less data. Therefore adding MODIS scene as a random intercept results in singular fits (which indicates an overly complex model), thus we cannot include this in the winter models. We have changed the manuscript to account for these changes (lines 272-293).

L34/35: selection “at” the population/individual level? 

We corrected this (line 34). 

L36: Why “spatio-“ here? This line only references differences in temporal scales.

Steps (displacement over a certain time period) inherently cover both space and time because the animals move in space and time. We categorize our steps in terms of time (1, 5, and 10 days) as is standard for SSFs, but these steps increase in time and distance traveled. Therefore we kept the word spatio-temporal, also because to interpret our results it was important to think about both the spatial and the temporal aspects of this analysis. Specifically, the spatial aspects play a role again in the discussion in the section ‘The role of landscape configuration and random searches.’ Here we essentially argue that because the spatial component has not been considered before, SSFs might not work in all landscapes or with all data types. 

L67: Why “interact to”? Are plant characteristics interacting, or is it simply that plant characteristics influence herbivore selection?

We changed the sentence to be more precise (lines 66-68):

 “For ungulates the forage maturation hypothesis (FMH) describes how plant characteristics and herbivore foraging constraints might interact to influence herbivores’ selection for vegetation.”

L77: “To work for” isn’t very clear. I suggest recasting this sentence, or simply cutting the first five words of the sentence.

We re-worded this part (lines 77 – 79):

L90, 126, 206: “Mongolian gazelles”

We corrected this. 

L113/114: References?

We added references for this statement (lines 114-115).

L177: I suggest changing this sentence a bit. This wording could mean that there are no open plains or rolling hills.

This is true, we changed the ordering in this sentence to improve clarity (lines 184-185). 

L187: Individuals cannot adapt, as this is a result of natural selection across generations. To make this sentence clearer, I suggest either using the term “acclimate”, or changing this to something along the lines of “gazelles must therefore be adapted to cope with a wide range of conditions”, depending on which meaning you wish to portray.

We changed the sentence to (lines 193-194): 

“Mongolian gazelles must therefore be able to tolerate a wide range of conditions across the year, and accordingly we split our analysis into two timeframes.” 

L190-191: NDVI is a value, not snow cover. How was snow cover determined from the NDVI layers? I assume based on the snow “flag”/layer of MODIS NDVI, but I think this should be explained. To be clear, this was when no pixels in the entire study area had a snow cover flag?

Thank-you for bringing this to our attention. We now have a section called “defining seasons” (lines 234 – 241) placed after the two MODIS data sets are introduced, so that our procedure is clearer. We use the snow cover data set to determine when there was snow during our study period and this is now explicitly stated (line 235-237). 

L209: Should this be “rarefied” instead of summarized?

We believe our original word choice is more accurate and simple. 

L236-237: Why did you crop MCPs? Was there a biological reason to exclude points outside Mongolia? What proportion of points did this represent?

Thank-you for pointing this out, we now realize this is not clear to readers not familiar with Mongolia. There have been double fences along the borders between Mongolia-China and Mongolia-Russia for nearly 30 years and so gazelles cannot travel outside the border, therefore this habitat is not available to them and cannot be considered in the analysis. We now emphasize this in the study area part (lines 186 – 188) and added a reminder about this for clarification in lines 251-251. In addition, the map of our study area which we added (Fig 1) shows how points remain within Mongolia and might make this point clearer. Points that appear just outside the border are gazelles stuck between these two fence lines. 

L277: “and snow”?

We corrected this (line 295). 

L293: “allowed us to sample”

We corrected this (line 311). 

L304-306: This is confusing. This describes the procedure for the winter models, yet the example is from a gazelle in spring? Obviously these are related, but should this be temporal scale instead of spatial (the spatial scale is changing because the temporal one is).

Thank-you, we missed this discrepancy and now changed spring to winter in the example (lines 324-325). 

L308: Typo in “therefore”.

We corrected this (line 328). 

L348: “At” the population level?

We corrected this (line 366). 

Table 1: The AIC Weight for growing season doesn’t make sense. For the top model to be 0.66 and the 2nd highest to be <0.001, then there has to be a lot of other models with small (but not insignificant) weights to make up the difference, Unless these are not the two top models?

The reviewer is indeed pointing to a mistake in the previous version, we accidently pulled the weight from a different comparison. We have corrected the table and now weights sum to 1. 

L391: I don’t follow this. Weren’t linear and quadratic the only shapes tested? So isn’t this statement guaranteed to be the case?

This is in contrast to the RSF. Based on the RSF we would expect that a quadratic model would be the best fit for all individuals, but it was not. So while these were the only shapes tested, it was a surprise that selection followed a linear shape in some gazelles and a quadratic shape in others. 

In Fig. 4 we can see that selecting for above average NDVI can happen two ways: by selecting for maximum NDVI (best fit model would be linear) and by selecting for intermediate NDVI (best fit would be a quadratic model). These are two different selection patterns. In the RSF the quadratic model was the best fit. In the SSFs we see that this quadratic model is not the best fit for all individuals. 

To avoid confusion, we now remind readers of the quadratic shape of the RSF and to compare Figure 2 and 4 (lines 415-417). 

L392-395: I think this needs to be reworked. I’m guessing the first sentence is suggesting a variety of responses across individuals, with some exhibiting positive and some exhibiting negative linear selection, and some with the downward parabola. This should be more explicitly explained. Stating that gazelles selected for max, min, and intermediate levels sounds like they simply select everything (see also lines 495-497). Is the 2nd sentence true for individuals who selected an intermediate amount of snow cover?

We now see how this is confusing and so we fixed the wording to improve clarity (lines 418-420 and 528-529)

L452: Remove first “for” in “to detect for selection for forage”.

This was corrected (lines 484-485). 

L454, 610: Change “preference” to selection. Preference typically is used to describe a situation where selection is quantified when availability is equal (e.g., a cafeteria trial).

We went through the manuscript and replaced preference with selection in these two instances and several others. 

L457-459: This is a very short disjointed paragraph with really no discussion per se.

We now combined the discussion of the winter results with the previous paragraph (lines 488-492). 

L460: This seems early in the Discussion to be conjecturing about differences between your study and others. I suggest moving this to near the end of the discussion.

In our study we compare selection at the population vs. the individual level. Therefore, we think this paragraph should remain in the section discussing the population level results to avoid confusion later in the discussion, but we now emphasized that these comparisons are only to the population level results in the beginning of the paragraph (line 493). 

L469: Citation?

We added citations (line 503). 

L574: What is meant by “important”? Important for what? For these to be classified as animal personalities?

We adjusted this sentence for clarity (lines 608-610).

Fig. 2: The overall resolution of this figure is quite low. The two shades of blue are also quite difficult to tell apart, especially in bars with only a single color.

The figure resolution is due to the PLOS ONE system. We ran all images through the PACE system as requested to meet journal standards, but the journal’s system reduced image quality in the PDF for speed and size. 

We changed the colors in the graphs to contrast better (Figs. 3 and 6 and S4 Fig). 

Fig. 4: Define the horizontal dashed line in the caption and legend and make it a different line type from the vertical line for snow cover in 2014/15.

This is now a light grey line and appropriately labeled in the legend and figure caption (Fig 5 and S3, S7, S8 Figs). 

Response to Reviewer #2: 

The manuscript “Resource selection of a nomadic ungulate in a dynamic landscape” addresses interesting topic of evaluating resource selection of nomadic Mongolian gazelle. The study provides an important contribution to this topic by addressing selection at multiple spatio-temporal scales using a solid sample size of observational data and appropriate modelling techniques. The strengths of the paper are the strong conceptual framework used to understand nomadic spatial behaviour in ungulates, which is currently understudied. And the complex evaluation of such behaviour across multiple spatial and temporal scales will result in this paper being important. The weakness, which I’m sure you are aware, is the use of proxies, such as NDVI for forage, and snow cover for snow depth. However, this is overshadowed to me by the importance and the rarity of studies on habitat selection of nomadic ungulates in particular, and the methods you used which are appropriate and rigorous. Finally, I think a short statement in the discussion regarding possible changes in behavior of nomadic Mongolian gazelle with climate change could help to improve the visibility of the paper. The study is generally well-perceived, concisely written and well organized. Thus, my concerns are relatively minor and mostly address unclear passages in the text.

Altogether I recommend only minor revisions of the manuscript.

We are happy to hear that the reviewer sees the relevance of our study and appreciates the efforts we made to conduct a rigorous analysis. We appreciate the comments made which have helped us improve the clarity of our text. We would love to be able to say more about the effects of climate change on gazelles. Climate change might affect important environmental conditions such as snow depth and vegetation composition, among other things, which gazelles would need to respond to. Unfortunately, we do not have sufficiently detailed environmental data to conduct such analyses. Yet we added a sentence to the discussion to highlight the importance of creating such data sets to enable research on climate change impacts in the future (lines 636-638):

“Better proxies for vegetation quality and snow depth data in combination with temperature data would be particularly valuable to examine responses of Mongolian gazelle to climate change.” 

L 56 – “That which is known” - consider re-phrasing

We re-phrased this sentence (line 56)

L 56 – “seasons” instead of “season”?

This was corrected (line 56). 

L 64 – … we used Mongolian gazelles… - use either "as a model species" or "for this case study"...

We change the wording to “…we used Mongolian gazelles for this case study.” (line 64)

L 77 – 79 – this is a bit confusing... FMH works for both, residents and migrants... both can select for intermediate forage biomass during whole growing season... try re-phrasing

Lines 77-79 were edited for clarity, according to this comment and Reviewer 1’s comment.

L 98 – unclear... it seems like you mean "trade-offs in selection between habitat types"

We edited this sentence for clarity (lines 98-101)

L 107 – 108 – I suggest improving, resp. re-phrasing this sentence

This sentence was reworked (108-109). 

L 114 – 115 – ..."to conclude"... instead of …”to arrive at an overall conclusion”…

We keep our original phrasing because “overall” was added to emphasize the fact that these studies combined results of many individuals and just focus on the overall conclusion instead of what the individuals do. 

L 126 – 130 – I think this description is more suitable for Methods/Study area description

We feel it is advantageous to introduce the Mongolian gazelle at this point so that our hypotheses have context and are easier to understand. 

L 140 - …”in a fission-fussion manner”… - Im not sure what this term supposed to mean within this context.

Thank you for pointing this out. Fission-fusion is a concept of herd dynamics, we have changed the wording for clarity and also cited a paper (Couzin and Laidre 2009) that describes the term in more detail (145-147).

L 162 – 163 – sounds like Methods

We agree, however we would prefer to keep the wording as is in this early stage of the manuscript to avoid any confusion later as to the use of NDVI and snow data in the different seasons. It also allows us to formulate our hypotheses more precisely later on. 

L 186 – 196 – I dont think this should go to Study area. This is the setting of the temporal scale of your study, it should be under description of the RSFs.

We created a new section labeled “Defining Seasons” to address this comment and related concerns by Reviewer 1 (lines 233-241). The seasons apply to both the RSFs and the SSFs. 

L 207 – 208 – How long were animals tracked? min - max months, average +-SD... How many points were collected from 1 animal? min - max, average +-SD

Thanks for pointing this out, this information is indeed important. Therefore we added the range, mean, and standard deviation of how long gazelles were tracked in the text (lines 207-208) and we also present more detailed information in supplementary table 1. 

L 222 – This sentence would need a citation.

Thank-you for point out that we do not show where this number comes from. Since we changed our methods we removed this sentence, but similar statements come later on. These numbers comes from our data and we now back up this statement (lines 307-309) with a histogram of 1, 5, and 10 day step lengths in supplementary figure 1. 

L 233 – ..."we generated"... instead of …”we had to generate”…

This was corrected (line 248). 

L 245 – I suggest re-phrasing the beginning of the sentence...

We changed this sentence (lines 260– 261)

L 292 – missing space - ..."and 10 days..."

This was corrected (line 311). 

L 293 – ..."allowed us"... instead of …”lets us”…

This was corrected (line 311). 

L 322 – 326 – Id say this is already Results...

Good point, we moved this and the corresponding text for the RSFs to the results section (lines 364-366 and 403-407). 

L 398 – By switching selection strategies, do you mean switching between linear and quadratic selection? And also, do you mean selection for NDVI or snow?

This was no very explicit so we added text to clarify this (lines 423-428)

L 454 – ... selection for higher NDVI... instead of ... preference for higher NDVI...

We went through the manuscript and replaced preference with selection.  

References 

Olson KA, Fuller TK, Mueller T, Murray MG, Nicolson C, Odonkhuu D, et al. Annual movements of Mongolian gazelles: Nomads in the Eastern Steppe. J Arid Environ. 2010 Nov;74(11):1435–42.

Mueller T, Olson KA, Dressler G, Leimgruber P, Fuller TK, Nicolson C, et al. How landscape dynamics link individual- to population-level movement patterns: a multispecies comparison of ungulate relocation data. Glob Ecol Biogeogr. 2011 Sep 1;20(5):683–94.

Fleming CH, Calabrese JM, Mueller T, Olson KA, Leimgruber P, Fagan WF. From Fine-Scale Foraging to Home Ranges: A Semivariance Approach to Identifying Movement Modes across Spatiotemporal Scales. Am Nat. 2014 May;183(5):E154–67.

Nandintsetseg D, Bracis C, Olson KA, Böhning‐Gaese K, Calabrese JM, Chimeddorj B, et al. Challenges in the conservation of wide-ranging nomadic species. J Appl Ecol. 2019;56(8):1916–26.

Nandintsetseg D, Bracis C, Leimgruber P, Kaczensky P, Buuveibaatar B, Lkhagvasuren B, et al. Variability in nomadism: environmental gradients modulate the movement behaviors of dryland ungulates. Ecosphere. 2019 Nov 1;10(11):e02924.

Bastille-Rousseau G, Potts JR, Yackulic CB, Frair JL, Ellington EH, Blake S. Flexible characterization of animal movement pattern using net squared displacement and a latent state model. Mov Ecol. 2016 Dec;4(1):15. 

Cagnacci F, Focardi S, Ghisla A, van Moorter B, Gurarie E, Heurich M, et al. How many routes lead to migration? Comparison of methods to assess and characterize migratory movements. J Anim Ecol. 2015;15. 

Singh NJ, Allen AM, Ericsson G. Quantifying Migration Behaviour Using Net Squared Displacement Approach: Clarifications and Caveats. Hays G, editor. PLOS ONE. 2016 Mar 3;11(3):e0149594. 

Bunnefeld N, Börger L, van Moorter B, Rolandsen CM, Dettki H, Solberg EJ, et al. A model-driven approach to quantify migration patterns: individual, regional and yearly differences: Quantifying migration patterns. J Anim Ecol. 2011 Mar;80(2):466–76.

Barbet‐Massin M, Jiguet F, Albert CH, Thuiller W. Selecting pseudo-absences for species distribution models: how, where and how many? Methods Ecol Evol. 2012;3(2):327–38.

Reinking AK, Smith KT, Mong TW, Read MJ, Beck JL. Across scales, pronghorn select sagebrush, avoid fences, and show negative responses to anthropogenic features in winter. Ecosphere [Internet]. 2019 May [cited 2021 Jan 20];10(5). Available from: https://onlinelibrary.wiley.com/doi/abs/10.1002/ecs2.2722

Lamont BG, Monteith KL, Merkle JA, Mong TW, Albeke SE, Hayes MM, et al. Multi-scale habitat selection of elk in response to beetle-killed forest. J Wildl Manag. 2019;83(3):679–93. 

Jakes AF, DeCesare NJ, Jones PF, Gates CC, Story SJ, Olimb SK, et al. Multi-scale habitat assessment of pronghorn migration routes. Grignolio S, editor. PLOS ONE. 2020 Dec 4;15(12):e0241042.

Couzin ID, Laidre ME. Fission–fusion populations. Curr Biol. 2009 Aug 11;19(15):R633–5.

---

## [Editor Report · Decision Letter 1]

27 Jan 2021

Resource selection of a nomadic ungulate in a dynamic landscape

PONE-D-20-30912R1

Dear Dr. Stratmann,

We’re pleased to inform you that your manuscript has been judged scientifically suitable for publication and will be formally accepted for publication once it meets all outstanding technical requirements.

Kind regards,

Floyd W Weckerly

Academic Editor

PLOS ONE

Additional Editor Comments (optional):

There was one revised sentence I think should be as originally written (ln 554-555).

As written:

This raises the question of what search strategies can be used when a landscape that has large, homogeneous patches.

Originally written:

This raises the question of what search strategies can be used when a landscape has large, homogeneous patches.
---

## [Editor Report · Acceptance letter]

3 Feb 2021

PONE-D-20-30912R1 

Resource selection of a nomadic ungulate in a dynamic landscape 

Dear Dr. Stratmann:

I'm pleased to inform you that your manuscript has been deemed suitable for publication in PLOS ONE. Congratulations! Your manuscript is now with our production department. 

Kind regards, 

on behalf of

Dr. Floyd W Weckerly 

Academic Editor

PLOS ONE